# Antibiotic-induced microbiome depletion alters metabolic homeostasis by affecting gut signaling and colonic metabolism

Amir Zarrinpar [1,2,3,4], Amandine Chaix [1], Zhenjiang Z. Xu[3,5,6], Max W. Chang [7,9], Clarisse A. Marotz[3,5], Alan Saghatelian[8], Rob Knight[3,5,6] & Satchidananda Panda[1]

Antibiotic-induced microbiome depletion (AIMD) has been used frequently to study the role of the gut microbiome in pathological conditions. However, unlike germ-free mice, the effects of AIMD on host metabolism remain incompletely understood. Here we show the effects of AIMD to elucidate its effects on gut homeostasis, luminal signaling, and metabolism. We demonstrate that AIMD, which decreases luminal Firmicutes and Bacteroidetes species, decreases baseline serum glucose levels, reduces glucose surge in a tolerance test, and improves insulin sensitivity without altering adiposity. These changes occur in the setting of decreased luminal short-chain fatty acids (SCFAs), especially butyrate, and the secondary bile acid pool, which affects whole-body bile acid metabolism. In mice, AIMD alters cecal gene expression and gut glucagon-like peptide 1 signaling. Extensive tissue remodeling and decreased availability of SCFAs shift colonocyte metabolism toward glucose utilization. We suggest that AIMD alters glucose homeostasis by potentially shifting colonocyte energy utilization from SCFAs to glucose.

[1] Regulatory Biology Laboratory, The Salk Institute, 10010 N. Torrey Pines Road, La Jolla, CA 92037, USA. [2] Division of Gastroenterology, University of California, San Diego, 9500 Gilman Drive, MC 0983, La Jolla, CA 92093-0983, USA. [3] Center for Microbiome Innovation, University of California, San Diego, 9500 Gilman Drive, MC 0436, La Jolla, CA 92093-0436, USA. [4] Division of Gastroenterology, VA San Diego Health Systems, 3350 La Jolla Village Drive, MC 9111D, La Jolla, CA 92161, USA. [5] Department of Pediatrics, University of California, San Diego, 9500 Gilman Drive, MC 0763, La Jolla, CA 92093-0763, USA. [6] Department of Computer Science and Engineering, University of California, San Diego, 9500 Gilman Drive, La Jolla, CA 92093, USA. [7] Integrative Genomics and Bioinformatics Core, The Salk Institute, 10010 N. Torrey Pines Road, La Jolla, CA 92037, USA. [8] Protein Biology Laboratory, The Salk Institute, 10010 N. Torrey Pines Road, La Jolla, CA 92037, USA. [9] Present address: Division of Endocrinology, University of California, San Diego, 9500 Gilman Drive #0640, La Jolla, CA 92093-0640, USA. These authors contributed equally: Amir Zarrinpar, Amandine Chaix. Correspondence and requests for materials should be addressed to A.Z. (email: azarrinp@ucsd.edu) or to S.P. (email: panda@salk.edu)

Recently, antibiotic-induced microbiome-depleted (AIMD) mice, where luminal bacteria are reduced with the administration of four or more broad-spectrum antibiotics through gavage or drinking water, have been used in conjunction with, or sometimes instead of, germ-free (GF) mice to investigate the role of the gut microbiome in some pathological conditions[1–3]. The metabolic effects of the absence of microbiota in GF mice have been well characterized—for example, change in the luminal and serum bile acid (BA) pool, increased incretin release, decreased short-chain fatty acid (SCFA) availability, and altered glucose homeostasis[4–6]. However, the effects of AIMD on the metabolic homeostasis of mice is unknown. Host transcriptomic studies in the gut of GF and AIMD mice show gene expression patterns that are distinct, suggesting that their metabolic physiology may be different[7]. With a better understanding of the effects of AIMD on host physiology, AIMD mice can be more effectively used in experiments investigating the role of the gut microbiome.

The consequences of low-dose antibiotic treatment aimed at inducing dysbiosis by decreasing diversity of the microbiome have been well documented[8–10]. These protocols have led to hyperglycemia, increased adiposity, and insulin resistance. However, the effects of an acute and dramatic depletion of the microbiome, as in AIMD, on metabolic homeostasis are not yet well characterized and can have paradoxical effects. For example, intestinal microbes are responsible for additional calorie extraction through the fermentation of indigestible carbohydrate into SCFAs[11]. Thus, a depleted microbiome could decrease the number of calories available for absorption and induce a hypoglycemic state similar to what is described in GF mice[5]. In fact, antibiotic-treated mice have decreased luminal SCFAs[4]. However, a microbiome that is deficient in butyrate-producing microbiota is associated with type 2 diabetes (T2D)[12,13]. There is some evidence that butyrate-producing microbiota and butyrate itself are protective against T2D[14,15]. Furthermore, by affecting luminal SCFAs, intestinal microbiota can modulate intestinal incretin excretion, and hence, apply another influence on glucose regulation[4].

AIMD can also affect metabolic homeostasis by altering the BA pool. BAs can affect host metabolism by signaling through the farnesoid X receptor (FXR) and G-protein-coupled BA receptor (TGR5)[6,16,17]. Intestinal microbiota can deconjugate BAs and convert primary BAs into secondary ones. Relative reduction of secondary BAs to primary BAs is associated with insulin resistance[18]. However, one of the most potent FXR antagonists is tauro-β-muricholic acid (TbMCA) which is deconjugated by bile salt hydrolases (BSHs) in intestinal bacteria. Suppression of BSH can lead to protection against obesity and improved insulin sensitivity[17]. Since AIMD reduces bacteria with BSH, it could also have a protective effect and improve insulin sensitivity.

Earlier mouse studies show that in cases of diet—or genetically induced obesity, AIMD can protect against dysmetabolism[19–22]. Furthermore, altered incretin signaling and BA signaling have been studied in GF mice[4,6]. However, AIMD mice are different from GF mice in three ways: (a) AIMD mice do not have complete depletion of their gut microbiome, (b) AIMD has normal development of immune and homeostatic regulatory system, and (c) antibiotics themselves have a direct effect on host homeostasis. If probiotics, prebiotics, or selective bacterial species targeting are used as therapeutic approaches for diseases, initial depletion of the microbiome by antibiotics is necessary to allow the reconstitution of the microbiota with selective species. Knowing the physiological effects of AIMD to normal metabolism is a prerequisite for implementing these potential therapies.

In these studies, we aim to investigate the effects of microbiome depletion on metabolism by performing AIMD in normal-chow fed, wild-type mice to understand its effects on host metabolism. The microbiome of AIMD mice is depleted of all major bacterial phyla except Proteobacteria, as evidenced by negative stool cultures, decrease in stool DNA, and engorgement of the cecum[19–23]. Our results show that AIMD mice have improved glycemic levels and increased insulin sensitivity despite having the same body weight, food consumption, and body composition. These changes in glucose homeostasis are associated with altered gut hormone release, serum and fecal BA pool, luminal SCFA profile, intestinal gene expression, and intestinal and hepatic glucose and BA metabolic pathways. AIMD reveals the deeply intertwined relationship between the gut, the microbiome, and glucose homeostasis.

## Results

**AIMD affects composition and function of the gut microbiome.** Twelve-week-old mice were given twice daily (i.e., every 12 h) oral gavage of ampicillin, vancomycin, neomycin, metronidazole, and amphotericin B (AIMD-treated group) or water (vehicle-treated group)[23]. Stool cultures prepared from vehicle-treated and AIMD-treated mice revealed far fewer colonies in the AIMD group (Fig. 1a and Supplementary Fig. 1A). The amount of stool-extracted DNA was nearly 20-fold lower in the AIMD mice compared to vehicle-treated mice (Supplementary Fig. 1B), suggesting depletion of all luminal organisms. Finally, AIMD resulted in a massive increase in cecal weight and size (Supplementary Fig. 1C, D), as well as increase in stool output (Supplementary Fig. 1E).

The pre-treatment microbiome of the mice was dominated by Firmicutes and Bacteroidetes species, as was the microbiome from the vehicle-treated mice (Fig. 1b, d, Supplementary Fig. 1F, and Supplementary Data 1). The microbiome of AIMD mice had far fewer sequences from the Firmicutes and Bacteroidetes phylum. Instead, the microbiome of AIMD mice had a compositional shift to Proteobacteria (Fig. 1c, d, Supplementary Fig. 1F, and Supplementary Data 1). These phylum level differences are significant based on an analysis of composition of microbiome (ANCOM) assessment, which uses log-ratio analysis to improve inferences from microbiome compositional survey data[24]. The pre-treatment and vehicle-treated mice shared a majority of their OTUs, whereas the AIMD mice, which had fewer OTUs, shared about a sixth of their OTUs with the other two conditions (Supplementary Fig. 1G)

Principal coordinate analysis of UniFrac distances[25] showed the microbiomes of pre-treatment and vehicle-treated mice were similar to each other, but the microbiome of AIMD mice was quite distinct (Fig. 1e). Phylogenetic diversity (PD) whole-tree (a measure of α-diversity) assessment of the three different groups of microbiomes showed that the pre-treatment condition had the greatest diversity, with a slight decrease in the vehicle-treated mice and significant decrease in AIMD-treated mice (Supplementary Fig. 1H, I). Another measure of diversity, UniFrac distance, showed that within-group distances for AIMD was significantly greater than that of pre-treatment and vehicle-treated conditions (Supplementary Fig. 1J). Between-group UniFrac distances (β-diversity measure) showed that the AIMD condition was equally distant from the pre-treatment and vehicle conditions and the pre-treatment and vehicle conditions were nearly as distant from each other as they were from members of their own group (Supplementary Fig. 1J).

These compositional and diversity changes induced by AIMD have functional implications. We used PICRUSt, a bioinformatics tool designed to predict metagenomic functional content from 16S ribosomal RNA (rRNA) surveys using known microbial genomes as reference, to assess potential functional differences

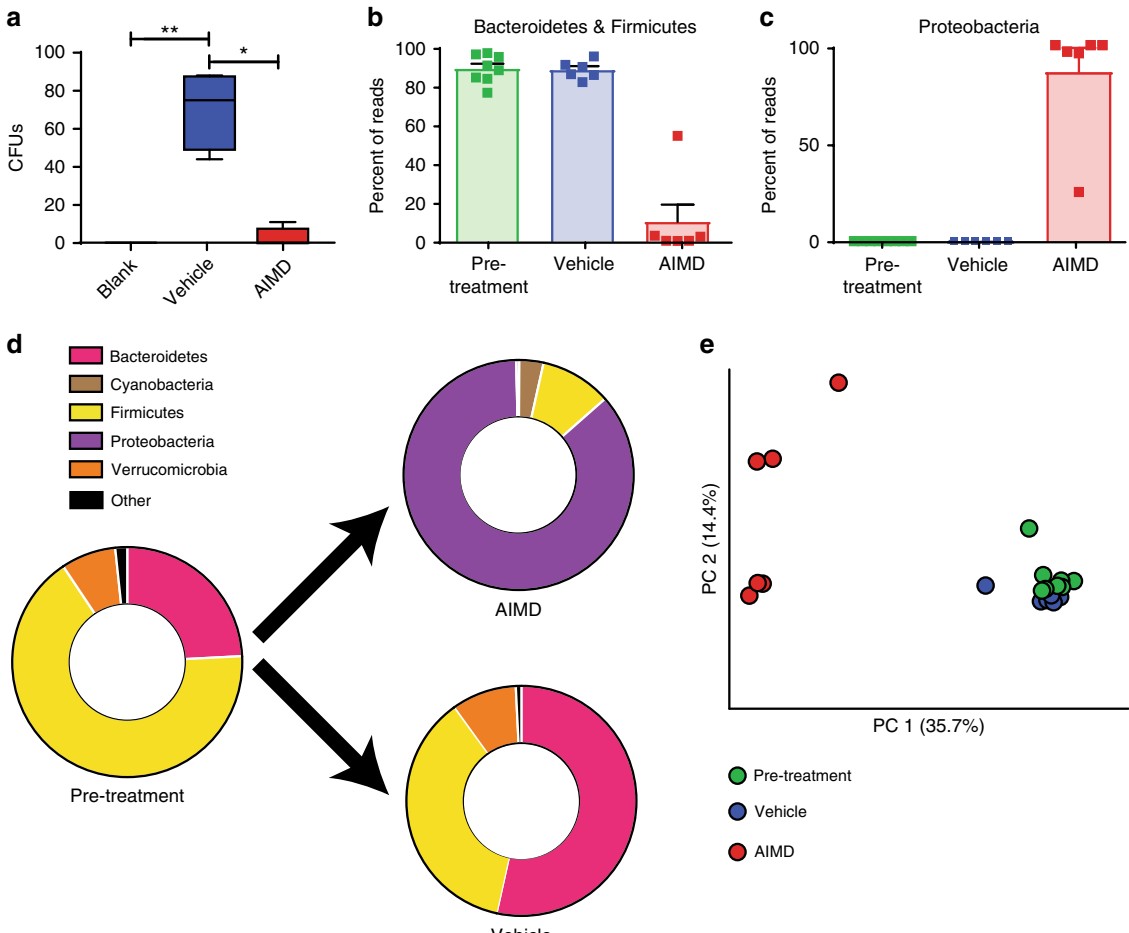

**Fig. 1** AIMD depletes the gut microbiome. **a** Stool cultures from AIMD mice yielded fewer colonies, $n = 5$/group. For box plot, center is mean, box is 25th to 75th percentile, and whiskers are 5th to 95th percentile. Kruskal–Wallis test, *$p < 0.05$, **$p < 0.01$. **b, c** 16S results show that AIMD mice had a decrease in OTUs detected from the Firmicutes and Bacteroidetes phyla, and an increase in Proteobacteria phyla, $n = 6$–8/group, mean percent abundance (with SEM). These differences were significant as assessed by ANCOM[24]. **d** 16S results showing the shifts in microbiome in each treatment condition. **e** Principal coordinate analysis of the gut microbiome, $n = 6$–8/group. Fecal specimens were collected approximately 2 weeks after intervention

between the microbiomes of AIMD-treated and vehicle-treated mice[26]. Many of the predicted functional differences were in metabolic pathways (Supplementary Data 2). The lipid metabolism pathway, including secondary BA biosynthesis, was significantly different between vehicle-treated and AIMD-treated mice (Supplementary Fig. 1K). In summary, AIMD in mice consuming a normal-chow diet-induced microbiome depletion, especially of Firmicutes and Bacteroidetes species, decreased microbial diversity, and altered the potential functional capabilities of the microbiome.

**AIMD improves glucose homeostasis**. We then assessed the effects of AIMD on glucose homeostasis. AIMD induced a lower fasting blood glucose level compared to vehicle-treated mice (Fig. 2a) after a 4-h fast or a 16-h fast. Glucose clearance was much faster in AIMD mice, with glucose levels lower in the AIMD group compared to the vehicle-treated controls (Fig. 2b). The area under the curve (AUC) above baseline for AIMD mice was less than half of that of vehicle-treated mice (Fig. 2c). An insulin tolerance test (ITT) showed that AIMD mice had increased sensitivity to insulin compared to vehicle-treated controls (Fig. 2d). However, fasting insulin levels was not

significantly different between AIMD-treated and vehicle-treated mice (Fig. 2e).

AIMD did not affect the weight of the mice (Fig. 2f and Supplementary Fig. 2A). The cumulative food consumption was also indistinguishable between the two groups (Fig. 2g and Supplementary Fig. 2B). The body composition of AIMD mice was similar to vehicle-treated mice, with the same amount of fat and lean mass, but there was a change in tissue that could not be characterized as being lean or fat (Fig. 2h and Supplementary Fig. 2C, D). This difference is likely due to changes in cecal content. Hence, AIMD resulted in a drop in fasting blood glucose, increased glucose tolerance, and increased insulin sensitivity despite unchanged serum insulin levels in the setting of unchanged mouse body weight, food consumption, and adiposity.

**AIMD alters luminal secondary metabolites**. AIMD can affect luminal secondary metabolites and gut signaling, which can affect whole-body glucose homeostasis. AIMD led to a complete disappearance of butyrate from the feces (Fig. 3a). AIMD mice also had a significant decrease in the fecal concentration of propionic acid when compared to vehicle-treated mice (Fig. 3a). Hence, AIMD changes the SCFA pool, most notably by decreasing butyrate to undetectable levels.

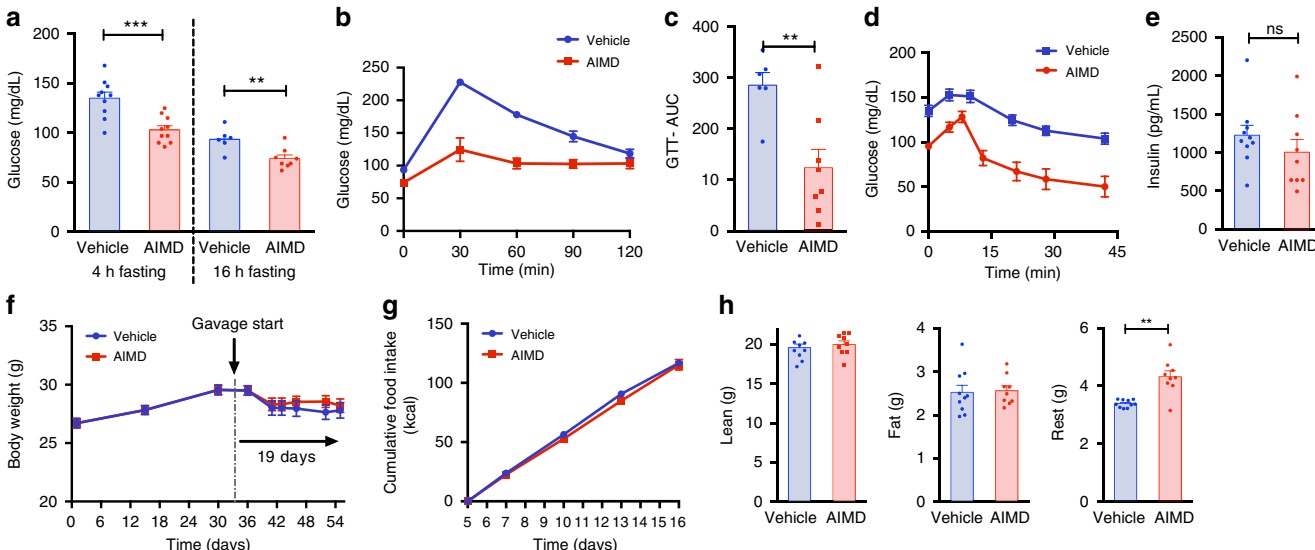

**Fig. 2** AIMD improves glucose tolerance and insulin sensitivity. **a** Blood glucose levels after 4 or 16 h of fasting. **b**, **c** Blood glucose concentration after IP injection of a glucose bolus (1 g/kg BW) (GTT) (**b**) and area under the curve quantification (**c**). **d** Blood glucose levels after IP injection of insulin (0.75 U/kg BW) (ITT). **e** Fasting serum insulin levels. **f** Body weight before and after the start of antibiotics gavage. **g** Cumulative food consumption (kcal) after acclimation to gavage. **h** Body composition as a percentage of total body weight. **f**–**h** were replicated in a separate experiment (Supplementary Figure 2). GTT and glucose measures were performed 3 weeks after intervention, and ITT was performed 4 weeks after intervention. Unpaired Student's $t$ test, $**p < 0.01$, $***p < 0.001$, ns nonsignificant. All error bars are SEM

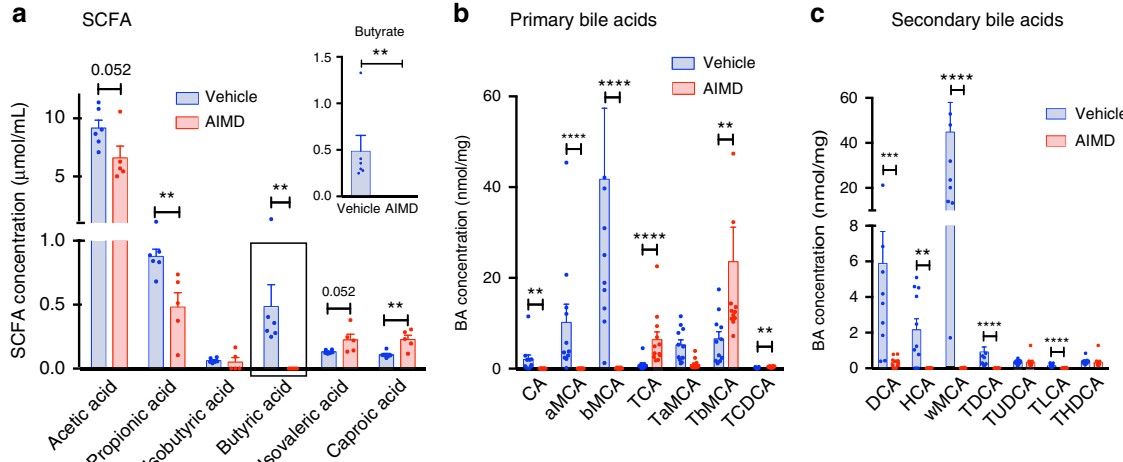

**Fig. 3** AIMD affects SCFA and BA profiles in the cecum. **a** Absolute quantification of the level of different SCFA in the cecal content. Insert: close-up of butyrate levels. Absolute quantification of primary (**b**) and secondary BAs (**c**) in the feces. CA cholic acid, aMCA α-muricholic acid, bMCA β-muricholic acid, TCA taurocholic acid, TaMCA tauro-α-muricholic acid, TbMCA tauro-β-muricholic acid, TCDCA taurochenodeoxycholic acid, DCA deoxycholic acid, HCA hyocholic acid, wMCA ω-muricholic acid, TDCA taurodeoxycholic acid, TUDCA tauroursodeoxycholic acid, TLCA taurolithocholic acid, THDCA taurohyodeoxycholic acid. Fecal samples were collected 2–3 weeks after intervention. Mann–Whitney $U$ test, $**p < 0.01$, $***p < 0.001$. All error bars are SEM

AIMD has substantial effects on the luminal BA pool. Fecal samples from AIMD mice had lower concentrations of many primary BAs (Fig. 3b and Supplementary Table 1) and secondary BAs (Fig. 3c and Supplementary Table 1). The ratio of primary to secondary fecal BAs is more than 20-fold higher in AIMD mice compared to vehicle-treated mice. Thus, AIMD is accompanied by significant changes in luminal SCFA and BAs. These changes confirm the functional predictions made by the PICRUSt analysis (Supplementary Fig. 1K).

**AIMD alters gut metabolic signaling.** Since changes in luminal secondary metabolites can alter intestinal homeostasis and gut

signaling, we assessed gut-released hormones and other metabolically important hormones. First, we measured serum levels of glucagon-like peptide 1 (GLP-1) and glucose-dependent insulinotropic peptide (GIP), which are incretins that can modulate serum glucose levels. AIMD mice had a nearly 18-fold higher total GLP-1 (Fig. 4a). In addition, blood collection on Ddp4 inhibitor-coated tubes revealed a higher fasted (Fig. 4b) and fed (Fig. 4b) active GLP-1. Although fasting insulin levels were not different, insulin significantly increased after re-feeding AIMD mice (Supplementary Fig. 3A). This suggests that GLP-1 potentiated insulin response after feeding.

In agreement with the observed increase in GLP-1, results from cecal transcriptome analysis (Fig. 4c) show that the expression

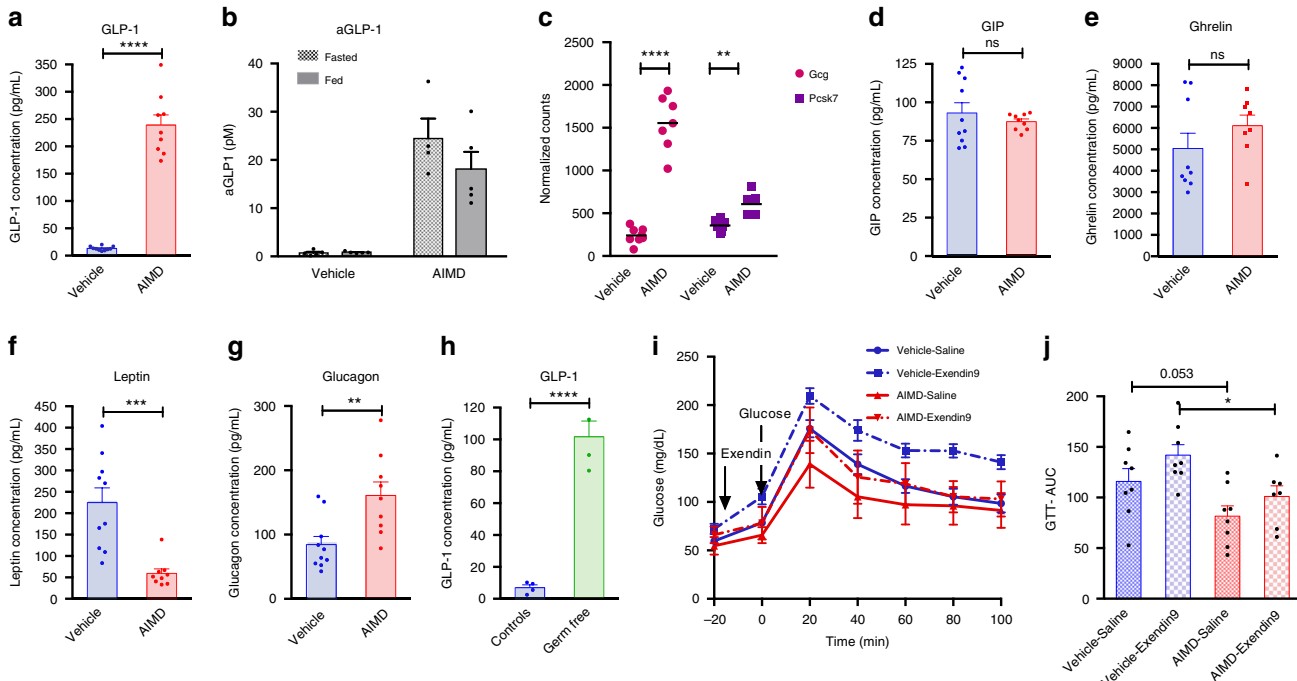

**Fig. 4** AIMD increases GLP-1 and affects other gut hormones. **a** Fasting serum level of total GLP-1 ($n = 9$–$10$/group). **b** Serum level of active GLP-1 after 16 h of fasting (fasted) or 15 min after an oral bolus of glucose (1 g/kg BW; fed). Blood was collected on Ddp4 inhibitor-coated tubes ($n = 5$/group). **c** Quantification by RNA-sequencing of Ggc and Pcsk7 mRNA expression in the cecum ($n = 7$–$8$/group). Adjusted $p$ value from negative binomial Wald test (corrected for multiple hypothesis testing with the Benjamini–Hochberg method), $*p < 0.05$, $**p < 0.01$, $***p < 0.001$, $****p < 0.0001$. **d**–**g** Serum endocrine panel—plasmatic levels of GIP (**d**), ghrelin (**e**), leptin (**f**), and glucagon (**g**) ($n = 9$–$10$/group). **h** Serum level of total GLP-1 in GF mice ($n = 4$/ group). **i**, **j** AIMD or vehicle mice were IP injected Exendin-9 (250 nmol/kg) or saline 20 min prior to receiving an oral bolus of glucose (1 g/kg BW) and glucose concentration was monitored over 100 min ($n = 8$/group). Quantification of the area under the curve (AUC, **j**). Serum collection and oGTT were performed at 1 week after intervention. Unless otherwise stated, Student's $t$ test, $*p < 0.05$, $**p < 0.01$, $***p < 0.001$, $****p < 0.0001$, ns nonsignificant. All error bars are SEM

level of the proglucagon gene (*Gcg*), which encodes GLP-1, was sevenfold higher in AIMD mice. In addition, the enzyme which converts preglucagon to GLP-1 and GLP-2, proprotein convertase subtilisin/kexin type 7 (Pcsk7) was elevated in AIMD mice. Dipeptidylpeptidase 4 (Dpp4), the enzyme that degrades GLP-1, was not different between AIMD-treated and vehicle-treated mice (Supplementary Fig. 3B). Finally, the expression of genes for other gut hormone that are co-secreted with GLP-1, including protein YY and cholecystokinin were also upregulated in AIMD mice (Supplementary Fig. 3B). GIP, on the other hand, was not affected by AIMD (Fig. 4d).

We also measured other metabolically important hormones. Ghrelin, a gut hormone released by the gut that affects appetite, was not significantly different between vehicle and AIMD mice (Fig. 4e). However, AIMD mice had significantly less leptin, another hormone that regulates appetite (Fig. 4f). Finally, AIMD mice had nearly twice the amount of glucagon in their serum (Fig. 4g).

We further assessed whether these changes in GI hormones are specific to AIMD mice or were also observed in GF mice. Since GF mice used for these experiments were older, housed in a different room, with a different vendor for their normal-chow diet, and were not receiving twice daily gavage as an intervention, we used conventionally raised brethren as controls. Similar to AIMD mice, GF mice had elevated levels of GLP-1 compared to their controls (Fig. 4h). Also similar to AIMD mice, GF mice did not have significant differences in their fasting serum insulin (Supplementary Fig. 3C) or GIP (Supplementary Fig. 3D). However, unlike AIMD mice, there was no significant difference

between GF mice and their controls in serum glucagon (Supplementary Fig. 3E) or serum leptin (Supplementary Fig. 3F). In addition, unlike AIMD mice, GF mice had lower production of ghrelin (Supplementary Fig. 3G).

We investigated whether the lower fasting blood glucose and the increased glucose clearance seen in GTT were caused by high levels of serum GLP-1. We used a high dose of Exendin-9 (Ex-9), a potent GLP-1 receptor antagonist to reverse the effects of GLP-1 in AIMD mice (Fig. 4i). If GLP-1 is the sole cause of hypoglycemia in AIMD mice, we would expect the GTT response to be similar between vehicle-treated and AIMD mice after Ex-9 injection. Although Ex-9 increased the glycemic surge in AIMD mice, it did not bring their oral glucose tolerate test (oGTT) response to the same level as that of vehicle-treated mice who also received the antagonist (Fig. 4j). In other words, the relative incretin effect of GLP-1 appears to be similar between AIMD-treated and vehicle-treated mice, which suggests that lower blood glucose in AIMD mice is not solely caused by excessive GLP-1.

**AIMD remodels cecal enterocyte transcriptome.** The effect of AIMD on luminal secondary metabolites and gut signaling suggests that this treatment is also affecting intestinal homeostasis. To further assess the effect of AIMD on the gut, we performed cecal transcriptome analysis. AIMD-treated and vehicle-treated mice cluster by treatment type in a principal component analysis analysis (Fig. 5a). The AIMD-treated and vehicle-treated groups had 6256 differentially expressed genes (Supplementary Fig. 4A and Supplementary Data 3). Ingenuity pathway analysis of these

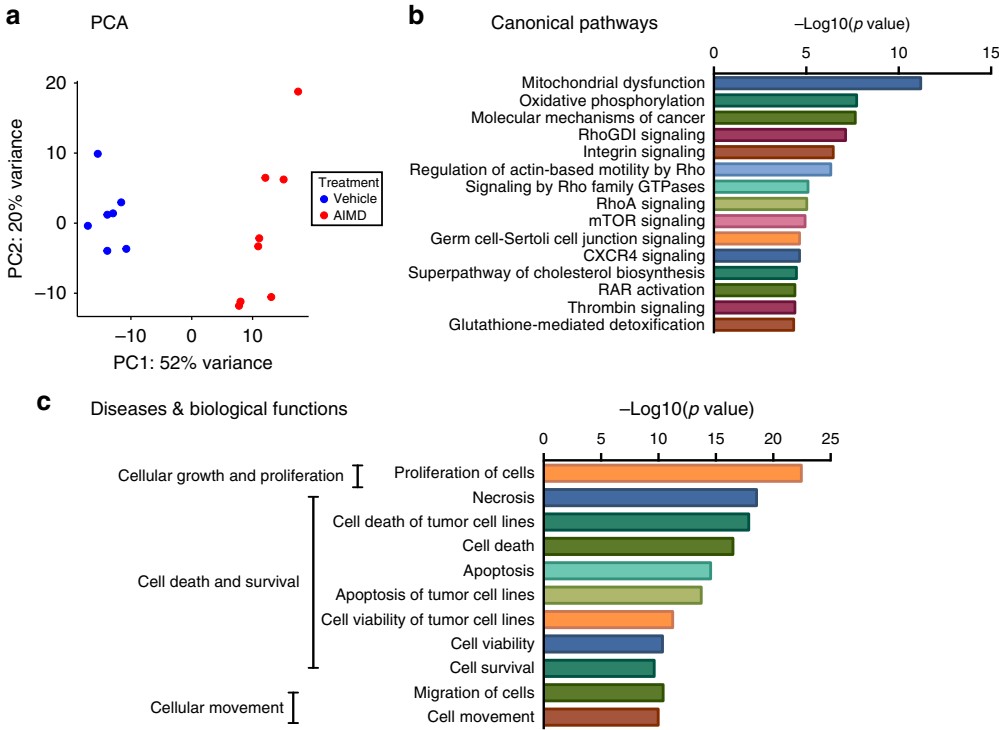

**Fig. 5** AIMD induces major changes in the transcriptome of the cecum. **a** PCA analysis of the cecum RNA-sequencing data ($n = 7/8/groups$). **b**, **c** Functional annotation of differentially expressed genes using IPA. The top 15 canonical pathways (**b**) and the top 11 biological functions (**c**) are shown

differentially expressed genes reveal that they are mainly involved in mitochondrial dysfunction and oxidative phosphorylation (Fig. 5b), which is consistent with the decrease in the main fuel of cecal enterocytes, butyrate (Fig. 3a). Furthermore, changes in cellular growth and proliferation, as well as cell death and survival pathways (Fig. 5c), suggest colonic remodeling, which is consistent with the larger cecum size observed in these mice (Supplementary Fig. 1C, D).

AIMD affected gut inflammatory signaling by activating both pro-inflammatory and anti-inflammatory cascades. For example, some pro-inflammatory transcripts were significantly higher in AIMD mice (Tlr4, Tnfrs1a; Supplementary Fig. 4B). However, other pro-inflammatory transcripts (e.g., interleukin-18 (IL-18); Supplementary Fig. 4B) were significantly lower in AIMD mice, as was an inhibitor of the pro-inflammatory cytokine IL-1 (IL1rm; Supplementary Fig. 4B). Hence, it is difficult to determine how AIMD affects gut inflammatory pathways through transcriptomics alone. In sum, changes in cecal gene expression suggest that AIMD not only induces gut proliferation and alters inflammatory signaling but also affects cecal enterocyte metabolism.

**AIMD shifts cecal enterocyte metabolism**. To better understand the altered glucose homeostasis associated with AIMD, we investigated the expression of genes involved in metabolic pathways. Most strikingly, AIMD leads to extensive reconfiguration of the cecal enterocytes' metabolism. The decrease in luminal SCFAs, particularly of butyric acid (Fig. 3a) and BAs (Fig. 3b, c), suggests difficulty in the absorption of fatty acids, the primary source of nutrients for enterocytes. Accordingly, compared to vehicle-treated mice, AIMD mice down-regulate genes involved in fatty acid metabolism (Fig. 6a). A schematic of transcriptional changes in the cecum is illustrated in Fig. 6b. There is a down-regulation of fatty acid receptors and transporters such as free

fatty acid receptor 2 (Ffar2/Gpr41) and fatty acid-binding proteins (e.g., Fabp2). SCFAs are imported into the mitochondria by carnitine palmitoyltransferases (Cpt1a, Cpt2) and processed by acyl-CoA synthetases (Acss1), both of which are downregulated in AIMD mice. Similarly, all the enzymes of fatty acid β-oxidation are downregulated in AIMD mice, including acyl-CoA dehydrogenases (Acadl, Acad8, Acad12), enoyl CoA hydratases (Echdc2), 2,4-dienoyl CoA reductase (Decr1, Decr2), hydroxyacyl-coenzyme A dehydrogenase (Hadh, Hadhb), and acetyl CoA acyltransferase (Acaa1a, Acaa1b). A parsimonious explanation of the consistent downregulation of β-oxidation genes is that the enterocytes are not using SCFAs for cellular metabolism.

Cecal enterocytes could be using SCFAs for energy storage through lipogenesis. However, cecal gene expression profiling again shows that this is not the case (Fig. 6a, b). AIMD mice have a downregulation of fatty acid synthase (Fasn) compared to vehicle-treated mice. There is also a downregulation of the stearoyl-CoA desaturase (Scd1) which elongates palmitate, the end product of fatty acid synthase complex, into oleate. Furthermore, the upregulation of 3-oxoacid CoA transferase (Oxct1) accompanied by a complementary downregulation in the genes involved in ketogenesis, including acyl-CoA thioesterase (Acot7) and HMG-CoA synthase (Hmgcs2), suggested increased utilization of ketone bodies as an energy source.

Cecal transcriptome analysis indicates that anaerobic glycolysis is upregulated in the enterocytes of AIMD mice (Fig. 6a, b). Out of the ten enzymes in anaerobic glycolysis, eight were upregulated in AIMD mice. AIMD have increased expression of hexokinase (Hk2), phosphofructokinase (Pfkp), aldolase B (Aldob), phosphoglyceratekinase (Pgk1), phosphoglycerate mutase (Pgam1), enolase (Eno1), pyruvate kinase (Pklr), and lactate dehydrogenase (Ldha). Appropriately, transcription for genes in the reverse pathway of glycolysis, gluconeogenesis (e.g., its rate-limiting

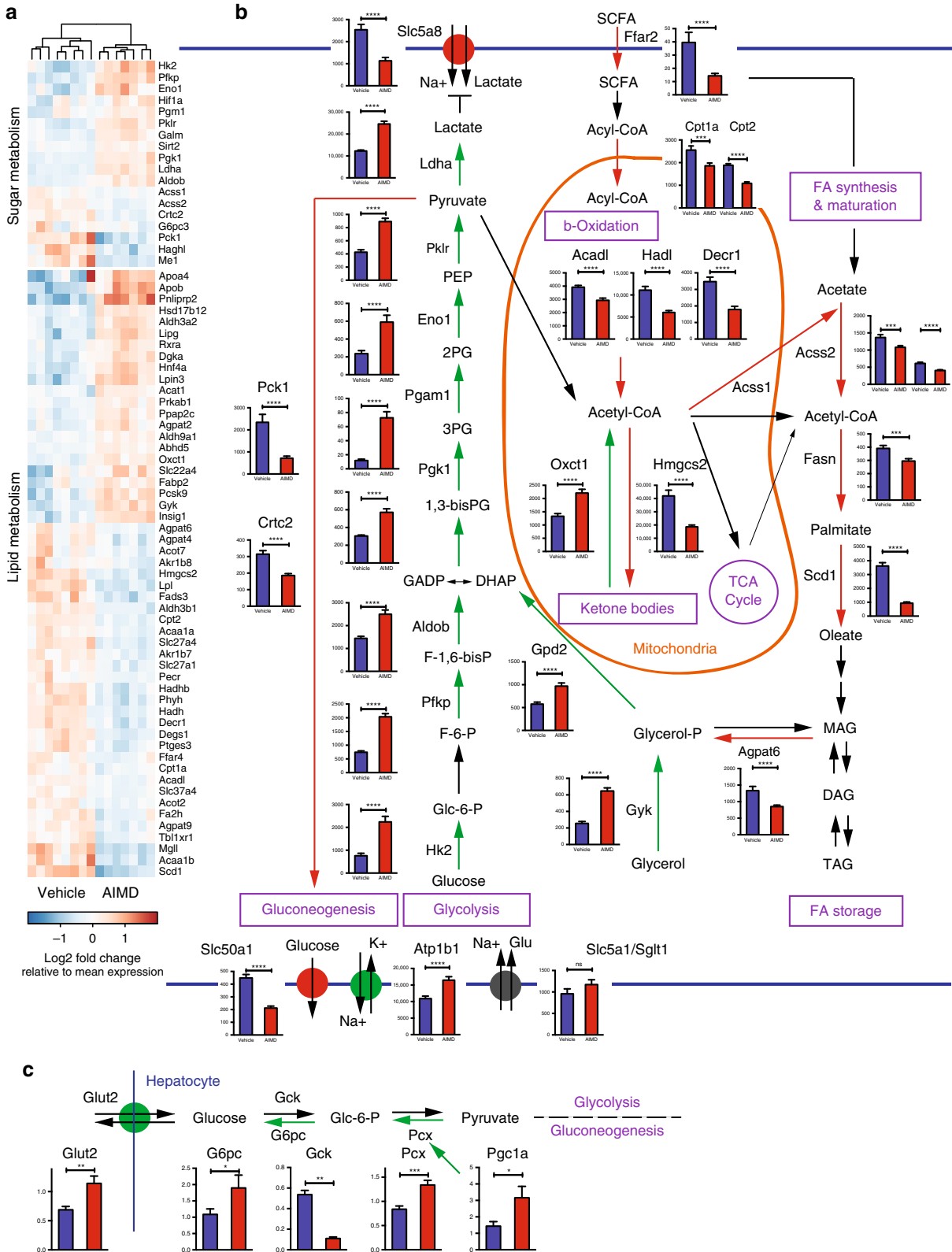

**Fig. 6** Anaerobic glycolysis in the cecum is upregulated in AIMD mice. **a** Heatmap of differentially expressed genes involved in sugar and lipid metabolism in the cecum of vehicle and AIMD mice as assessed by RNA-sequencing analysis ($n = 7$–8/group). **b** Mapping of metabolic genes onto specific pathways, including gluconeogenesis and glycolysis, β-oxidation, ketone body metabolism, TCA cycle, fatty acid synthesis, maturation and storage. Adjusted $p$ value from negative binomial Wald test (corrected for multiple hypothesis testing with the Benjamini–Hochberg method), $*p < 0.05$, $**p < 0.01$, $***p < 0.001$, $****p < 0.0001$. **c** Quantification by qRT-PCR of Glut2, Gck, G6pc, Pcx, and Pgc1a mRNA expression in the liver ($n = 8$/group). Tissue collected 4 weeks after intervention. Mann–Whitney $U$ test, $*p < 0.05$, $**p < 0.01$, $***p < 0.001$, ns nonsignificant. All error bars are SEM

enzyme phosphoenolpyruvate carboxykinase (Pck1) and its main transcriptional regulator, CREB, regulated transcription coactivator 2 (Crtc2)) are downregulated. Further analysis shows upregulation of genes transporting sugars to the glycolysis pathway (e.g., increased glycerol kinase (Gyk) and glycerol phosphate dehydrogenase (Gpd2)). These findings suggest that the primary source of energy for enterocytes is now glucose through anaerobic glycolysis, with less reliance on the mitochondria.

There are significant changes in the role of mitochondria in the metabolism of enterocytes. The proportion of total reads that map to mitochondrial chromosome M are reduced in AIMD mice, suggesting a reduced number of mitochondria (Supplementary Fig. 5A). Furthermore, mitochondrial cytochrome b and some subunits of cytochrome c oxidase such as Cox6a1 (complex IV) expression are significantly reduced in AIMD mice (Supplementary Fig. 5B). In agreement with increased utilization of glucose for anaerobic glycolysis as the main source of energy for enterocytes, expression of the sodium/potassium-transporting ATPase is upregulated (Atp1b1) (Fig. 6a, b). In addition, the sugar efflux transporter (Slc50a1) is downregulated, suggesting that the enterocytes are preventing glucose export.

These changes in intestinal glucose regulation can be responsible for the observed differences in whole-body glucose homeostasis in AIMD mice. Since the liver is the main regulator in glucose homeostasis, we characterized the hepatic expression of key glucose metabolic pathways using quantitative reverse transcription PCR (qRT-PCR). Unlike enterocytes, hepatic gluconeogenesis is upregulated with an increase in pyruvate carboxylase (Pcx) and its activator peroxisome proliferator-activated receptor gamma coactivator 1α (Pgc1a), as well as glucose-6-phosphatase (G6pc; Fig. 6c). Conversely, glucokinase, an enzyme in the glycolysis/reverse pathway, is downregulated. These results suggest that the liver in AIMD mice is actively producing more glucose. This is further confirmed by the upregulation of hepatic glucose transporter 2 (Glut2/Slc2a2), which is a facilitated transporter of glucose from the liver into the blood. There was no difference in hepatic histology (Supplementary Fig. 5C) or hepatic triglyceride load (Supplementary Fig. 5D) between AIMD-treated and vehicle-treated mice. Transcripts involved in lipid metabolism, such as Pparα (Supplementary Fig. 5E) and Rev-erbα (Supplementary Fig. 5E), are not affected by AIMD or secondary luminal metabolites.

In summary, examination of cecal transcriptomes reveals that proliferative enterocytes shift their metabolic nutritional needs away from SCFAs toward the utilization of glucose. The low amount of luminal SCFAs likely leads to a downregulation of fatty acid uptake, β-oxidation, and lipogenesis. Instead, there is an upregulation of anaerobic glycolysis and downregulation of gluconeogenesis and function of mitochondria. The upregulation of anaerobic glycolysis is similar to a Warburg effect in the setting of upregulated proliferation that correlates with high levels of GLP-1 growth factors. Furthermore, hepatic gene expression suggests that the liver responds to AIMD as though it is in starvation, upregulating the expression of genes in gluconeogenesis, while downregulating glycolysis genes.

**AIMD alters BA metabolism and potentiates GLP-1 response.** Since the luminal BA pool was quite different in AIMD compared to vehicle-treated mice, we examined the cecal enterocytes sequencing results for genes involved in BA processing (Fig. 7a). The expression of the primary transporter for intestinal BA uptake, Slc10a2/Ibat (also known as Asbt), is increased 16-fold in AIMD mice compared to vehicle-treated controls. In addition, organic solute transporter β (Slc51b/Ostb), a subunit of the Ost

basolateral transporter complex responsible for exporting BA from enterocytes to the enterohepatic circulation system, is increased 2.5-fold. Both transporters are activated by FXR, and the luminal BA pool contains both FXR agonists (e.g., taurocholic acid (TCA), cholic acid (CA)) and antagonists (e.g., TaMCA, TbMCA). FXR transcription is significantly increased in AIMD mice. In addition, FXR forms an obligate heterodimer with the retinoid X receptor (RXRa) which is also upregulated in AIMD mice. These changes suggest that FXR signaling in the gut is actively increasing the uptake of BAs from the lumen.

Since cecal transcriptome analysis indicates that there is an upregulation of BA reabsorption from the cecum, we assessed whether the altered luminal BA pool resulted in differences in the serum BA pool. Serum samples from AIMD mice have significantly higher amounts of common primary BAs (Fig. 7b and Supplementary Table 2). Serum secondary BAs remain higher in samples collected from vehicle-treated mice compared to those from AIMD mice (Fig. 7c and Supplementary Table 2). The ratio of primary to secondary serum BAs is more than tenfold higher in AIMD compared to vehicle-treated mice (Supplementary Fig. 6A). Consistent with observations from fecal BAs, CA and bMCA were significantly decreased in AIMD mice (Fig. 7b and Supplementary Table 2). Furthermore, there was nearly a twofold increase in TCA, a threefold increase in TbMCA, and a threefold increase in TCDCA (Fig. 7b and Supplementary Table 2). Despite altered serum and luminal BA pool in AIMD mice, there was no significant difference in serum cholesterol and triglyceride levels (Supplementary Fig. 6B, C).

Altered serum BA pool suggests altered BA metabolism in hepatocytes. We assessed the expression level of key enzymes by qRT-PCR in whole liver extracts of AIMD-treated and vehicle-treated mice (Fig. 7d). AIMD-induced serum BA pool changes coincide with increased expression of the primary basolateral membrane BA transporter, liver BA transporter (Slc10a1/Lbat). The primary BA increase did not affect the expression of FXR in hepatocytes, but did affect the expression of its main target and metabolic mediator, small heterodimer partner (Nr0b2/Shp). Shp regulates BA synthesis by inhibiting cholesterol 7 α-hydoxylase (Cyp7a1), the rate-limiting enzyme in de novo synthesis of BAs from cholesterol. As expected with downregulation of Shp, there is a nearly threefold increase in the expression of Cyp7a1 in hepatocytes.

Finally, we investigated whether the altered luminal BA pool contributed to the altered glucose homeostasis and increased GLP-1 observed in the AIMD mice. To assess this, we gavaged 12-week-old male wild-type C57BL/6 with TCA twice daily (i.e., every 12 h) for 4 days. TCA was selected since it was elevated in AIMD condition, whereas other BAs were not. TCA-treated mice had a significant increase in GLP-1 compared to vehicle-treated controls (Fig. 7E) and a decrease in fasting blood glucose level that trended toward significance (Supplementary Fig. 6D). Compared to vehicle-gavaged controls, glucagon, insulin, leptin, ghrelin, and GIP levels were unchanged in TCA-gavaged mice (Supplementary Fig. 6E–I).

In summary, in response to the lower luminal BA concentration, cecal enterocytes in AIMD mice increase the expression of FXR and upregulate BA transporters likely to conserve as much BAs as possible. The transport of primary/conjugated BAs that are FXR antagonists (e.g., TbMCA) into the serum in turn suppresses hepatic FXR targets, such as Shp, which allows Cyp7a1 to upregulate further BA synthesis. Finally, increasing luminal primary BAs by oral gavage with TCA elevates serum GLP-1, suggesting a potential mechanism by which altered luminal BAs in AIMD mice could lead to altered glucose homeostasis.

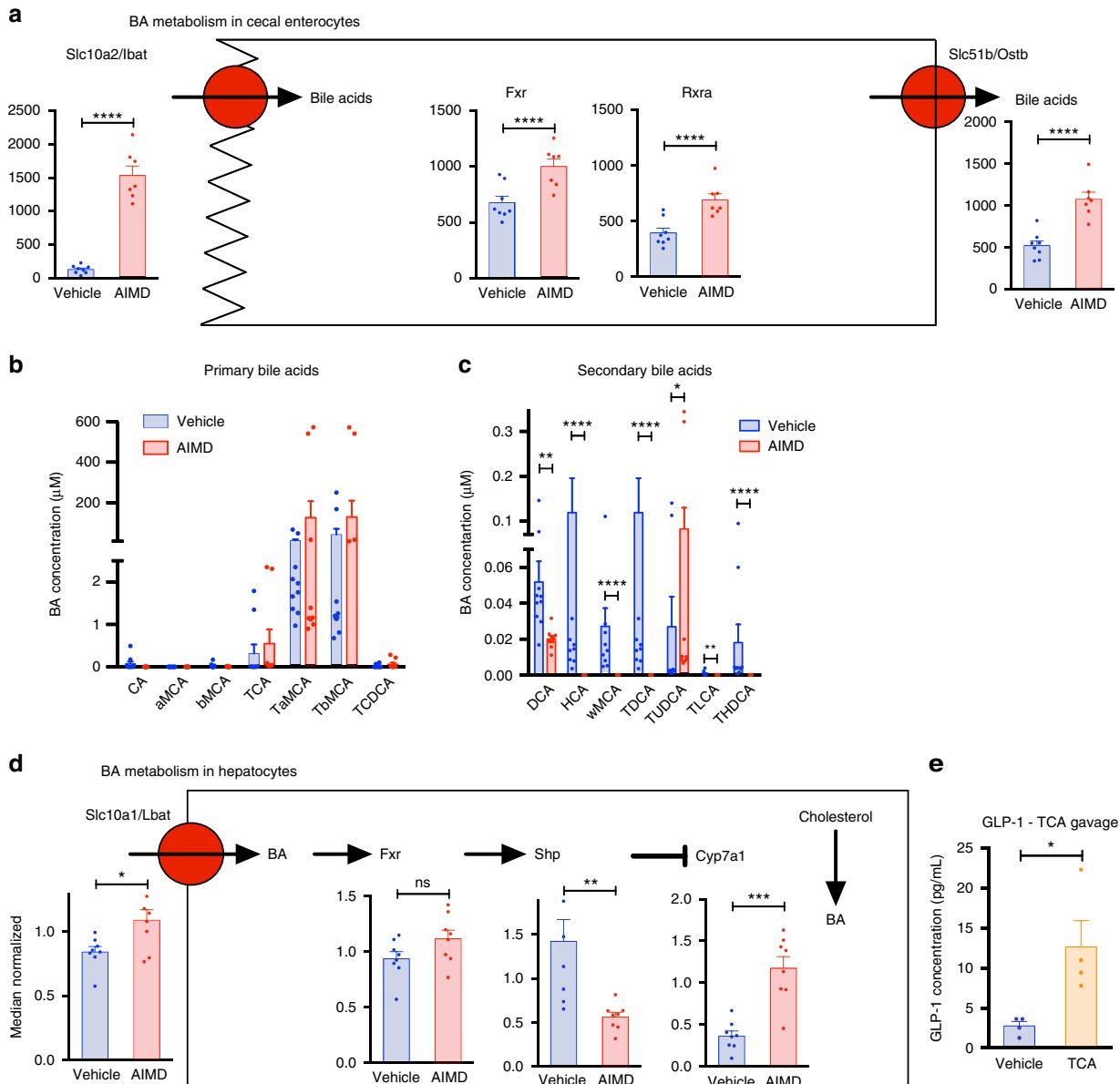

**Fig. 7** AIMD alters BA metabolism which can potentiate GLP-1 response. **a** Quantification by RNA-sequencing of Slc10a2, Fxr, Rxra, and Slc51b mRNA expression in the cecum ($n = 7$–8/group). Adjusted $p$ value from negative binomial Wald test (corrected for multiple hypothesis testing with the Benjamini–Hochberg method), *$p < 0.05$, **$p < 0.01$, ***$p < 0.001$, ****$p < 0.0001$. **b**, **c** Absolute quantification of primary (**b**) and secondary BAs (**c**) in the serum. Mann–Whitney $U$ test, *$p < 0.05$, **$p < 0.01$, ***$p < 0.001$, ****$p < 0.0001$. **d** Quantification by qRT-PCR of Slc10a1, Fxr, Shp, and Cyp7a1 mRNA expression in the liver ($n = 8$/group). Mann–Whitney $U$ test, *$p < 0.05$, **$p < 0.01$, ***$p < 0.001$, ns not significant. **e** Serum GLP-1 level after 4 days of bi-daily TCA gavage (400 mg/kg) ($n = 4$/group). Tissue and serum samples from AIMD studies collected at 4 weeks after intervention. Serum from TCA study was collected after 1 week intervention. Student's $t$ test, *$p < 0.05$. All error bars are SEM

## Discussion

AIMD induces major changes in luminal secondary metabolites including SCFAs and BAs that in turn affect gut signaling. By depriving the colon of its main fuel source, butyrate, the enterocytes appear to compensate by using glucose to meet their metabolic needs. This affects whole-body glucose homeostasis by lowering serum glucose, improving insulin sensitivity, and increasing hepatic gluconeogenesis. These changes occur in the setting of increased GLP-1 which likely results from increased luminal primary BAs, such as TCA. In addition to glucose utilization by the gut, AIMD causes the gut to selectively upregulate BA absorption. At the whole-body level, there is an increase in

serum BAs that are antagonists to FXR, which increases de novo synthesis of primary BAs in hepatocytes.

Our study confirms earlier studies that show that AIMD has a significant impact on glucose homeostasis. Treatment of both obese leptin-deficient (*ob/ob*) and diet-induced obesity (DIO) mice with norfloxacin and ampicillin for 2 weeks improved fasting blood glucose and oral glucose tolerance[20]. DIO mice had an increase in GLP-1 secretion in response to antibiotic treatment[21] and lower serum LPS[19], which could affect glucose homeostasis through alteration of the gut barrier function and decreased endotoxemia[27]. More recent studies show that, similar to GF mice, antibiotic treatment lowers blood glucose levels,

increases insulin sensitivity, and lowers oGTT in normal-chow-fed animals[22,28]. In addition, our results confirm previous studies that showed decreased luminal SCFAs in GF and antibiotic-treated mice affects Gcg expression[4].

Our study adds to this body of knowledge by demonstrating that chronic increased levels of GLP-1, a known intestinal growth factor[29], likely promotes colonic proliferation and remodeling and slows intestinal transit. The increased energy demand by the colon, in addition to the loss of its main fuel, butyrate, increases the gut's glucose utilization (i.e., anaerobic glycolysis). Previous studies which investigated the effects of antibiotic administration in GF mice and transplantation of antibiotic-resistant bacteria in GF mice show that both conditions decrease active mitochondria and inhibit mitochondrial gene expression, which could be contributing to increased epithelial cell death[7]. We have observed a similar phenomenon where mitochondrial number and gene expression are decreased in AIMD mice. This could also be contributing to increased anaerobic glycolysis observed in AIMD mice. Ultimately, this metabolic change in cecal enterocytes can make the gut a glucose sink, with increased glucose requirements that alter whole-body metabolic homeostasis. This may be a unique process to the colon, since in microbiome-depleted states, the jejunum, which does not use SCFAs as its primary source of energy, decreases glycolysis[30]. High levels of GLP-1, as in AIMD mice, likely slow intestinal transit[4,31], though the metabolic effects of this are incompletely understood. More recent studies of the microbiome of the proximal GI tract and gut barrier permeability independently play an important role in the host metabolic homeostasis[19,30,32]. Focused studies in these two areas may be necessary to appreciate the full effect of this intervention on host metabolic homeostasis.

AIMD induces decreased fasting serum leptin levels without a change in body adiposity. Besides body adiposity, a fed state (as determined with high circulating glucose) and increased serum insulin can both increase leptin release[33]. Although fasting insulin is similar between AIMD-treated and vehicle-treated mice, there is a significant difference between fasting and fed insulin in AIMD mice who are also hypoglycemic, both of which could explain decreased leptin in these mice. In addition, GLP-1 is known to affect satiety and high levels of GLP-1 may inhibit leptin release. Leptin is a known to be a potent stimulator of GLP-1 secretion from L cells[34,35]. Because high levels of serum GLP-1 is a potent appetite suppressor in our AIMD mice, decrease in serum leptin levels may be a response to help regulate food/energy intake. Hence, the relationship between GLP-1 on adipose tissue in general, and leptin release in particular, should be more closely investigated.

In addition, AIMD had an impact on BA physiology. Although both FXR agonists (e.g., TCA) and antagonists (e.g., TbMCA) are elevated in the luminal BA pool, FXR expression, and its downstream targets (e.g. Ostb, Ibap) were upregulated in the cecum. Furthermore, FXR activation in the gut suggests an increased uptake of BAs, which is confirmed by the serum BA pool which is more equivalent between AIMD-treated and vehicle-treated mice. However, gut uptake disproportionately increases serum TbMCA, a known FXR antagonist. In turn FXR activity is decreased in hepatocytes. This is confirmed by lower expression of Shp, the main target of FXR, and increased de novo BA synthesis by Cyp7a1. Although TGR5 gene expression levels were not different in the cecum of AIMD-treated and vehicle-treated mice, it is likely that AIMD-induced BA changes are also affecting TGR5 signaling which could also influence glucose homeostasis[36,37]. AIMD also increases TUDCA, a BA with known hepatoprotective properties that could also play a role in decreasing inflammation[38]. Hence, AIMD-induced changes in BA

physiology could likely affect host metabolism through various methods, most of which remain to be explored.

Studies using mouse models suggest that dysbiosis could contribute to T2D and metabolic diseases through multiple mechanisms, including alteration of luminal SCFAs, especially butyrate[11], the stool and serum BA pool[6,17], and the incretin signaling[39]. Furthermore, both obesity and T2D are associated with reduced microbial diversity in the gut lumen and/or feces[40]. Treating T2D with AIMD is unlikely to be tolerable to patients, nor acceptable to the medical community at large. In fact, short-term use of a single antibiotic did not have any metabolic effects[41]. Other studies investigating the effects of antibiotics on glucose homeostasis have focused on the long-term consequences and risk of T2D[42]. Nevertheless, AIMD mouse models are important for understanding the role of the gut microbiome in human glucose homeostasis. For example, current GLP-1 therapy for the treatment of T2D involves subcutaneous injection of GLP-1 receptor agonists or Dpp4 antagonists which is a barrier to their use. Since AIMD studies show that serum GLP-1 levels can be increased without an injection, understanding the mechanism by which the microbiome can affect incretin release can help find an oral agonist of incretin signaling.

Although AIMD mice may be preferable to GF mice in some experiments due to their relatively preserved immune function, normal development, and low cost, studies using AIMD should interpret their results cautiously. An acute depletion of the microbiome still has significant metabolic consequences. AIMD perturbs dynamic fluctuations of the gut microbiome driven by diet, genetics, and feeding rhythms, which are important for intestinal homeostasis and host metabolism[43,44]. Though it is unclear whether the metabolic consequences of AIMD are due to the depletion of the microbiome or administration of antibiotics, or combination thereof, it is likely that they will have a significant impact on other host physiological processes.

## Methods

**AIMD and TCA gavage**. The care and use of laboratory animals in our study was approved by the Salk Institute IACUC committee, and we have complied with all relevant ethical regulations. C57BL/6 mice (Jackson Laboratories) were housed in barrier facility and were housed 3–4 mice per cage. They were fed a normal-chow diet (LabDiet 5001). After an acclimation period, 12-week-old male mice were pseudorandomized into two groups. This pseudorandomization was based on the initial weight of the mice; in the end, the mean and standard deviation of the weight of the mice was the same between the two groups. One group of mice were given oral gavage of antibiotics (AIMD-treated group) or water (vehicle-treated group) every 12 h. The antibiotic cocktail was comprised of four antibiotics and one antifungal (ampicillin (100 mg/kg, Amresco Inc.), vancomycin (50 mg/kg, Amresco Inc.), metronidazole (100 mg/kg, MP Biomedicals), neomycin (100 mg/kg, Amresco Inc.), amphotericin B (1 mg/kg, Amresco Inc.)). Amphotericin B was added to prevent fungal overgrowth or opportunistic infections. The other group of mice received water. This protocol was maintained for different length of time (30, 19, and 13 days) leading to identical phenotypic results. This cocktail was made fresh every 36 h. Previous studies show that this method of antibiotic administration is the most effective in depleting the microbiome with minimal effects on the morbidity and mortality of the animal[23]. Experiments with once-daily dosing of all the antibiotics, or using each antibiotic in the cocktail individually did not induce hypoglycemia, or microbiome depletion which replicated previous findings[23].

For the TCA gavage experiment, 12-week-old male C57BL/6 mice (Jackson laboratories) were given oral gavage of TCA (400 mg/kg, 4 days, Sigma-Aldrich) every 12 h. TCA solution was prepared fresh daily.

**Stool culture measurement**. To assess microbiome depletion, fresh fecal samples were collected in pre-weighed tubes containing 1 mL phosphate-buffered saline (PBS) and their weights measured. Samples were thoroughly vortexed and diluted with PBS to 1:4. Fecal samples for AIMD-treated and vehicle-treated mice were collected 13 days after gavage initiation. Equal concentrations of stool diluent from AIMD-treated, vehicle-treated, and blanks were placed on LB agar plates and incubated for 7 days. One vehicle-treated sample that contained 1006 colonies was excluded from further analysis since it was an outlier.

**Glucose and ITTs**. Glucose tolerance assays were performed on fasted mice (16 h, paper bedding) by monitoring glucose levels after a glucose bolus (1 g/kg of body weight (BW)) by intraperitoneal (IP) injection or gavage as indicated 2 weeks after gavage initiation. For insulin tolerance, mice were fasted (16 h) and injected IP with insulin (0.5 U/kg BW). Glucose concentration was measured from the tail using a OneTouch Ultra glucometer. For the GLP-1 antagonist study, Ex-9 (250 nmol/kg body weight, Sigma-Aldrich) or 0.9% saline (vehicle control) was injected IP 20 min prior to the glucose bolus.

**Body weight and food consumption**. BW and food consumption were monitored weekly during the acclimation period and every 2–4 days after the start of the antibiotics/vehicle gavage.

**Body composition**. Whole-body composition was determined using the EchoMRI-100H analyzer according to the manufacturer's instruction prior to mice being euthanized (i.e., day 30 for the first cohort and day 19 for the second cohort).

**Stool weight**. Approximately 2 weeks after gavage initiation, individual animals were placed in clean cages for 24 h. All feces were collected and weighed.

**Cecum weight**. Mice were euthanized with carbon dioxide asphyxiation followed by decapitation. Each cecum was carefully dissected out and weight was determined using a precision scale. This experiment was performed in the second cohort (i.e., day 19 after gavage initiation).

**Germ-free mice and serum collection**. Germ-free C57BL/6 mice were bred and maintained in sterile semi-flexible isolators and screened for bacterial, viral, and fungal contamination[45]. Mice were fed autoclaved chow (LabDiet 5010). Conventionalized brethren in the same facility and diet were used as controls. For serum collection, germ-free mice were fasted for 6 h and blood was collected by cheek bleeding using a sterile lancet. Blood was centrifuged in BD Microtainer® at 6000 rpm for 20 min to isolate the serum.

**Short-chain fatty acids**. SCFA concentration in the feces was determined by gas chromatography/mass spectrometry by the AFNS Chromatography Facility at the University of Alberta (http://afns-labs.ualberta.ca/Central-Labs-Chromatography).

**Bile acids**. BA quantification was performed by the metabolomics core services at the University of Michigan from fecal samples collected from the first cohort of mice (approximately 3 weeks after gavage). BAs were extracted from feces (50 mg) using a two-step solvent extraction. Supernatants were combined, dried, and resuspended for liquid chromatography-mass spectrometry separation by reversed phase liquid chromatography. Results were quantified using standard curves of authentic standards. Data were averaged across samples.

**Histology**. Sections (6 μm) of formalin-fixed liver tissue attained from the first cohort of mice (i.e., 30 days after gavage initiation) were stained with hematoxylin and eosin.

**Liver triglycerides**. Liver powder obtained from the first cohort of mice (i.e., 30 days after gavage initiation) was homogenized in isopropanol and triglyceride concentration was measured using an enzymatic assay (Triglycerides LiquiColor, Stanbio) as per the manufacturer's instructions. Data are shown normalized to liver weight.

**Blood collection and serum biomarker measurements**. Blood was collected from fasted (4–16 h) or re-fed animals (15 min to 1 h). Re-feeding was performed by injecting a bolus of glucose (1 g/kg of BW) IP or by oral gavage as indicated. For active GLP-1 measurement, blood was collected on Dpp4 inhibitor-coated tubes (BD P700). Serum glucose, triglycerides, and cholesterol levels were assessed using Infinity reagents (Thermo Scientific) as per the manufacturer's instructions. Serum endocrine panel (ghrelin, GIP, GLP-1, glucagon, insulin, leptin) was performed using the Bio-Plex Pro Mouse Diabetes 8-Plex Assay (Bio-Rad) as per the manufacturer's protocol.

**Cecum transcriptomics**. Ceca were collected from the first cohort of mice (i.e., 30 days after gavage initiation), emptied of their content, and flash-frozen. Libraries were prepped using Illumina's TruSeq Stranded mRNA HT Kit according to the manufacturer's instructions. In brief, total 1 μg RNA was poly-A selected, fragmented by metal-ion hydrolysis and then converted to complementary DNA (cDNA) using SuperScript II. The cDNA was then end-repaired, adenylated, and ligated with Illumina sequencing adapters. Finally, the libraries were enriched by PCR amplification. Libraries were pulled into groups of 12 to a lane and sequenced using an Illumina HiSeq 2500 with 50-bp single-read chemistry.

Sequencing reads were mapped to the GRCm38 genome with STAR[46]. Gene-level read counts were generated using featureCounts[47] and GENCODE gene annotation. Analysis of differential expression was carried out using DESeq2[48], with a two-factor design that accounted for time of sacrifice and antibiotic treatment. Statistical significance was assessed using a negative binomial Wald test, then corrected for multiple hypothesis testing with the Benjamini–Hochberg method.

Principal component analysis was performed using the 500 genes with highest variance across all samples. A heatmap was generated by selecting differentially expressed genes associated with sugar or lipid metabolism. A regularized logarithm transformation[48] was applied to the gene-level read counts, which were then mean-centered and hierarchically clustered for visualization.

**Quantitative RT-PCR**. RNA and cDNA were prepared and qRT-PCR performed[49]. Absolute transcript expression was calculated using the standard curve method (using three technical replicates), normalized to 18S RNA, actin and RPL10 expression, and finally median normalized group wise ($n = 8$/group).

**Microbiome composition by 16S rRNA amplicon analysis**. Fecal samples from the same time point were collected from individually housed mice the day before gavage treatment initiation. Likewise, before the end of the experiment fecal samples were collected from individually housed AIMD-treated and vehicle-treated mice. Stool were frozen until DNA collection. The stool pellets were then weighed and resuspended in PBS and DNA was extracted using QIAamp DNA Stool Mini Kit (Qiagen), precipitated, and washed with ethanol. Resulting DNA was resuspended in TE. 16S rRNA gene sequence tags, corresponding to the hypervariable V3–V4 region, were generated using the MiSeq sequencing platform. The full-length primer sequences are[50]: 16S amplicon PCR forward primer = 5′-TCGTCGGCAGCGTCAGATGTGTATAAGAGACAGCCTACGGGNGGCWGCAG; 16S amplicon PCR reverse primer = 5′-GTCTCGTGGGCTCGGAGATGTGTATAAGAGACAGGACTACHVGGGTATCTAATCC.

The 9,584,677 total raw reads from Illumina MiSeq were quality controlled using QIIME v1.9.1 with default parameters, followed by closed-reference OTU clustering at 97% similarity level against the GreenGenes database (v13_8), resulting 4,344,060 reads in total (average of 164,103 for NA, 164,989 for aNA, and 296,188 for pre-treatment). After rarefying the samples to the same sequencing depth of 20,000 reads per sample, α-diversity, β-diversity, and differential OTU abundance analyses were performed on the rarefied OTU table with QIIME and ANCOM.

**PICRUSt analysis**. The functional profile of KEGG Orthology (KO) for each sample was predicted from 16S rRNA amplicon data with PICRUSt. The predicted KO abundances were collapsed to level 3 by grouping them into a higher level of functional categorization.

**Statistics**. Most experiments had an $n$ of 6–8 mice per group. Our previous experience has shown that this number is sufficient to notice a 20% change between groups[43,49,51]. AIMD-treated and vehicle-treated mice (two groups) comparisons were done with unpaired $t$ tests (serum measurements) or Mann–Whitney $U$ test (SCFA or BA measurements). Three-way comparisons (pre-treatment vs. AIMD vs. vehicle) were done with Kruskal–Wallis test. To address the compositional problem inherent in sequencing data sets (i.e., 16S microbiome differences), we performed ANCOM analysis[24] to statistically test the differential abundance between pre-treatment/vehicle group and AMID group. Statistical significance of differential gene expression from RNA-sequencing data was assessed using a negative binomial Wald test, and then corrected for multiple hypothesis testing with the Benjamini–Hochberg method. No blinding was performed for these experiments.

**Data availability**. 16S sequencing data is available in the European Bioinformatics Institute (EBI) database with the ascension number ERP109010 and the Qiita database under study ID 11785. Annotated 16S OTUs are located in Supplementary Data 1. PICRUSt results from our 16S sequencing results is located in Supplementary Data 2. Cecal RNA-seq data is available at NCBI Gene Expression Omnibus (GEO) with the accession number GSE114818. Supplementary Data 3 contains the annotated sequences and differential expression of genes in our conditions.

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

## Acknowledgements

We are indebted to Janelle Ayres and Michelle Lee for providing us serum from germ-free mice and Luis Rios for assistance with animal experiments. A.Z. received support from NIH K08 DK102902, NIH R03 DK114536, AASLD Liver Scholar Award, American Heart Association Beginning Grant-in-Aid (16BGIA27760160) and the American Gastroenterological Association Microbiome Junior Investigator Research Award. A.C. received salary support from an American Diabetes Association Mentor-Based Post-doctoral Fellowship (7-12-MN-64) and AHA grant 18CDA34110292. S.P. received support from NIH R01 DK115214. The Leona M. and Harry B. Helmsley Charitable Trust grant #2012-PG-MED002, Glenn Foundation, and NIH center grants P50 GM085764, P30 CA014195, P30 EY019005, KL2 TR00099, and R24 DK080506 supported various aspects of the study. Metabolomic Core Services were supported by NIH U24 DK097153 to the University of Michigan.

## Author contributions

A.Z., A.C., and S.P. conceptualized and designed the studies. A.Z., A.C., and C.A.M. carried out the wet lab experiments. A.Z. and A.C. analyzed all the results. A.Z. and A.C. wrote the manuscript. Z.Z.X. and R.K. provided additional analysis of the microbiome

data. A.S. provided additional analysis of the metabolomics results. M.W.C. performed cecal transcriptome bioinformatics analysis.

## Additional information

**Competing interests:** The authors declare no competing interests.

