## [Peer Review File · Nature Communications]

Reviewers' comments:

Reviewer #1 (Remarks to the Author):

The work by Zarrinpar et al. characterized metabolic alterations evoked by a cocktail of antibiotics that results in dramatic reduction of gut microbiota in mice. The rationale is that a better characterization of this model is needed so that it can be used more in studies addressing the role of microbiota. Examining cecum, the authors came to the conclusion that there is higher glucose utilization in this part of the gut after antibiotics as a result of deficit in short chain fatty acids. Importantly, some of these results are not entirely novel (e.g. ref 3, 6) while others do not agree with previous studies (ref. 3, PMID 22722618, PMID: 24412651). Importantly, it is not clear which of the described effects are truly microbiota-dependent and which ones are context-dependent, i.e specific antibiotic mix, the baseline microbiota and the resulting antibiotic-resistant microbiota. Thus the effects might not be generalizable. Further, comparison to GF is very limited, making it hard to conclude which effects are related to microbiota absence vs. antibiotic toxicity vs. antibiotic-resistant microbes.

In addition, the central claims regarding GLP-1, insulin are at odds with previous studies (ref. 3) and this fact has not been addressed.

Of note, the employed model of administration of Abx is very inconvenient and different from hundreds of other studies employing antibiotic-treated mice as a surrogate of germfree. Thus, it is doubtful that the results of this study that differ from other reports will provide generalizable insights.

Overall, the authors are investigating a very important topic. However, several problems in experimental design as well as failure to address differences between their and previously reported results lessen the potential importance of this study.

These and other comments are described in detail below.

-GLP-1 incretin effect and insulin changes in microbiota-depleted mice do not agree with a study by Wichmann et al., 2013 (ref. 3) who extensively studied this relationship and concluded that the role of colonic GLP-1 was primarily on intestinal transit in the large intestine and not an incretin effect because the excursion of glucose was similar in both groups after exendin9. They showed SCFA decrease related to GLP-1 increase in microbiota-depleted mice evoking a feedback mechanism of slower gut transit. Neither intestinal transit was examined nor was the discrepancy addressed in the current manuscript. - Glp1r KO mice should be used to test the contribution GLP1 as has been done in ref. 3.

-Depending on antibiotic cocktail, the administration route and the baseline microbiota, the effects might be different in different protocols of AIMD and would not necessarily agree with the complete microbiota depletion (germfree). Although gammaproteobacteria in general would be expected to dominate, particular species might differ, which can result in different outcomes.

-increased glycolysis has been reported in the gut after microbiota introduction into germfree mice (PMID 22722618). This also seems to contradict the results of the current manuscript bringing up again the possibility of confounding effects of antibiotics or some other variables.

-the main claim of the gut as a 'glucose sink' is not sufficiently corroborated. It could be directly compared the measurements of tissue glucose levels between control, AIMD and germfree mice.

-how can downregulation of gut gluconeogenesis suggested here be reconciled with studies showing that intestinal gluconeogenesis is important for improved systemic glucose tolerance (work by Mithieux lab and others)?

- direct effect of antibiotics on mouse cells cannot be excluded. For example, authors observed a discrepancy between AIMD and GF mice (figs. 4, S4). Can those be related to mitochondrial and

other metabolic deficiencies intestinal epithelial or other cells as previously reported (ref.6)? Comparison to GF is quite limited in this study. In particular, in the discussion there is a statement that switch to anaerobic glycolysis cannot be attributed to mitochondrial deficit. Why? With the defect in mitochondrial oxidative phosphorylation, cells are trying to compensate energetic demands by boosting glycolysis.

-This work represents a rather extensive description of an AIMD model, which would be not feasible to implement for most labs (gavage 2 times a day every day for 30 days!). More importantly, it is not justified why such protocol is needed, considering enormous number of studies that have been using simpler protocols (drinking water) for more than a decade.

-regarding the microbiome analysis (fig. 1): why showing together Bacteroidetes & Firmicutes? Which genera/species of proteobacteria were detected in AIMD mice?

-What if corrected by decreased overall absolute abundance, would amount of proteobacteria differ between groups, i.e. absolute amount per g of stool?

-cecal weight (and likely other parts of distal gut) is increased in almost 2 g in AIMD mice but body weight did not differ from controls. Something is missing. Decreased leptin levels (fig4) indicate that there might be less fat.

- regarding fig. 3: not all primary BA were lower, some were higher. All abbreviations of figs. 3B,C need to be explained. fig 3C is not the best representation. Why make fig. S3? It can be combined with main fig 3.

-regarding transcriptomic data: it would be important to know if AIMD transcriptomes are very different from GF or not in order to address the concern of the effects of antibiotics on mammalian tissues. Also, whether immune alterations can be attributed to antibiotic-resistant microbe outgrowth.

-levels of fasting insulin might be actually different in fig S4a at fasted state with means ~ 1.5 vs ~ 0.5 .

-was the effect specific to TCA? TbmCA was also increased in AMD mice.

-Would the phenotype be the same after oral GTT?

-several lipid metabolism genes show increased expression in AIMD mice (fig.6a), e.g ApoB, ApoA4, Hsd17b12 etc. What's the interpretation of this?

- using t-test might not be appropriate if the data did not have normal distributions, which might be the case at several instances.

-abstract needs to mention analyses of microbiota

-it is not stated how many times the experiments were repeated and although stated, data are not shown for different days of duration of the protocol giving "identical" results.

Reviewer #2 (Remarks to the Author):

In this manuscript, the authors describe the effect of depleting the microbiome using antibiotics on metabolic homeostasis. This is an important topic, regarding the numerous studies demonstrating a contribution of the gut microbiota in metabolic diseases as well as the increasing number of projects involving gut microbiota transplants. Using a mice model, they show that Antibiotic

induced microbiome depletion (AIMD) decreases baseline serum glucose levels, improves glucose tolerance and insulin sensitivity without altering food intake or body adiposity. Additional analyses including assays of short chain fatty acids, bile acids and gut hormones as well as cecal transcriptome analysis lead to the hypothesis that AIMD shifts the gut metabolism toward glucose utilization as an explanation of the observed phenotype.

The findings are interesting and certainly very well described. Overall, the paper is well written and the results represent significant additions to previously-published studies. Nevertheless, several important concerns can be inferred:

Major comments:

. The stool cultures presented in the paper (Fig 1A and S1a) have not been done properly. Bacterial population in mouse feces varies between 10^{11} to 10^{12} bacteria/g feces. Even if a part of the microbiota is not culturable, a minimum of 10^{10} (10 billions) cfu/g of feces is needed to have a representative idea of the bacterial population. Were the cultures done anaerobically? If not, this is completely useless as the great majority of the gut bacteria are strict anaerobic microbes. Moreover LB is definitely not a medium for culturing gut bacteria. Therefore, the results presented in Fig 1A and S1A does not tell anything about the bacterial population in the mice gut and must be deleted. Proper stool cultures must be done anaerobically using different culture media and lead to at least 10^{10} cfu/g feces.

Similarly, the authors assayed the concentration of DNA/g of stool. This is not clear why they did not measure bacterial population using real-time PCR using this DNA. This would tell which percentage of the bacteria has been depleted by AIMD.

. Experimental procedures indicate that AIMD protocol was maintained for different length of time (13 to 30 days) leading to identical results. However, it is not clear which duration was used for the represented results.

. Oral gavage every 12 hours must create a stress in the mice. It would have been interesting to measure the parameters that can be assayed on living animals (glycemia, insulinemia, glucose tolerance, fecal metabolites...) before the start of AIMD to evaluate effects of gavaging, independently of the effect of the antibiotics.

. Gene expression of gut hormones have been analyzed in cecum. While cecal transcriptome analysis is globally relevant, the main expression site of some of these hormones is not cecum. Therefore, it would be more relevant to analyse expression of ghrelin in the stomach, and of CCK and GIP in the duodenum.

. The methods section about the microbial community analysis is uncomplete. Specifically the number of raw sequences, the number of chimeras, filtered sequences due to other quality reasons, rarefaction of data, number of minimum sequences from each sample etc. should be spelled out as part of supplementary material. The number and description of OTUs shared by Pre-treatment vs AIMD vs vehicle must be specified. Also Beta diversity should be indicated.

. It has been reported that "metabolic endotoxemia", defined as a moderate increase in the concentration of bacterial lipopolysaccharide (LPS) in the plasma, is a triggering factor linking the gut microbiota to metabolic disorders. Plasma LPS concentration should be assayed to decipher if the effect of AIMD on glucose homeostasis is due, at least in part, to reduced endotoxemia. Similarly, altered gut barrier has been implicated in the gut bacteria induced metabolic disorders. Gut permeability analysis as well as expression of tight junction proteins would be useful to determine if AIMD influence gut barrier function.

. Several information on animal experiment procedures are lacking: How were sacrificed the

animals ? How many mice per cage...? Also information showing that guidelines for the care and use of laboratory animals were respected and approved by an ethical committee must be provided.

Minor comments:

- . line 70: in humans too !
- . line 337: Serum BA are expressed nmol/mg ? mg of what ? or is it ml ?
- . line 397: reference must be provided
- .line 722: Bacteroidetes instead of Bacteriodetes (also in Figure 1 legend)
- . Line 421: It is stated "AIMD studies can help fin an oral agonist of incretin signaling". It is not clear what the authors mean
- . Table S3 shows that AIMD increases some of the primary BA (TCA, TbMCA) and decreases some other (CA, aMCA, TaMCA). This must be discussed
- . The authors compare some of their results to what has been obtained in GF animals. GF animals are widely used for gut microbiota transplants experiments leading to the demonstration of a causative relationship between the gut microbiota and host phenotype. Do the authors believe that AIMD is a better model than GF for microbiota transplant ? This must be argued and discussed.
- . Isovaleric and caproic acids are increased in AIMD. Please comment and propose an explanation.
- . AIMD mice have less blood leptin while similar fat mass than vehicle-treated mice. This must be discussed
- . Globally the effects of specific BAs are poorly described. Also no information is given on TGR5 expression and its potential involvement in the observed effects. Please see some reviews (Chávez-Talavera O et al, Bile Acid Control of Metabolism and Inflammation in Obesity, Type 2 Diabetes, Dyslipidemia, and Nonalcoholic Fatty Liver Disease, Gastroenterology. 2017;152(7):1679-1694 ; Gérard P. Metabolism of Cholesterol and Bile Acids by the Gut Microbiota. Pathogens. 2014; 3(1):14-24....) and comment your results accordingly.
- . In vehicle treated mice, Bacteroidetes increases and Firmicutes decreases compared to pre-treatment. Is it significant ? Effect of stress due to gavage ? Please comment.
- . Fig S1F: Two different groups seem to appear regarding composition of the microbiota in Pre-treatment mice. It is important to consider possible "cage effects" in rodent studies, especially when examining the microbiome. It is known that microbiota of cage mates are correlated, and in some cases, cage effects have been shown to exceed the effects of exposures under study. One point the authors do not address is the housing of mice, and potential housing effects. Please indicate how the mice were housed during the normal course of the experiment and if the differences seen in this figure is related to a cage effect.
- . Fig 2 legend: cumulative instead of cunultaive
- . FigS3 is not informative. It may let think that primary BA increased and secondary BA decreased in AIMD while the ratio is very in AIMD because secondary BA are extremely low.
- . Fig4 and S4: controls in blue must be defined
- . Fig 7: TUDCA is increased in AIMD serum. This should be highlighted and discussed, especially because this BA is relevant in term of health and disease and may explain some of the effects of the gut bacteria on host metabolism (see Llopis et al, Gut, 2016, 65(5):830-9 as an example)

Reviewer #3 (Remarks to the Author):

In this manuscript, Dr. Zarrinpar and his/her colleague reported that antibiotic induced microbiome depletion alters metabolic homeostasis by affecting gut signaling and colonic metabolism. This research showed that AIMD improves insulin sensitivity without altering food intake or body adiposity thorough decrease of butyrate and secondary BA.

This experimental data and the interpretation are insufficient and the results do not present logical representation. They insist that AIMD did not alter food intake (Fig2G) and body adiposity (Fig2F, H, FigS2ACD). However, Food intake of AIMD tends to be low compared with vehicle. Moreover, although the cecum weight of AIMD is more than 2g (FigS1D), both body weights are same. This is discrepancy. Therefore, their AIMD phenotype can be explained by the difference of adiposity. I think that GLP-1 high concentration despite SCFAs and BA low concentration in AIMD is also due to this reason.

Other

Fig3A- What bacteria related on butyrate decrease by AIMD?

Fig4 and Fig4S- Why does value of all hormones differ between Vehicle and Controls?

Fig5- They should perform cecum histological analysis and cell assay as well.

Fig7E- They should examine aMCA, bMCA (Vehicle: high) and TbMCA (AIMD: high) in addition to TCA (AIMD: high).

RESPONSE TO REVIEWERS' COMMENTS

Reviewer #1 (Remarks to the Author):

The work by Zarrinpar et al. characterized metabolic alterations evoked by a cocktail of antibiotics that results in dramatic reduction of gut microbiota in mice. The rationale is that a better characterization of this model is needed so that it can be used more in studies addressing the role of microbiota. Examining cecum, the authors came to the conclusion that there is higher glucose utilization in this part of the gut after antibiotics as a result of deficit in short chain fatty acids. Importantly, some of these results are not entirely novel (e.g. ref 3, 6) while others do not agree with previous studies (ref. 3, PMID 22722618, PMID: 24412651). Importantly, it is not clear which of the described effects are truly microbiota-dependent and which ones are context-dependent, i.e specific antibiotic mix, the baseline microbiota and the resulting antibiotic-resistant microbiota. Thus the effects might not be generalizable. Further, comparison to GF is very limited, making it hard to conclude which effects are related to microbiota absence vs. antibiotic toxicity vs. antibiotic-resistant microbes. In addition, the central claims regarding GLP-1, insulin are at odds with previous studies (ref. 3) and this fact has not been addressed.

Of note, the employed model of administration of Abx is very inconvenient and different from hundreds of other studies employing antibiotic-treated mice as a surrogate of germfree. Thus, it is doubtful that the results of this study that differ from other reports will provide generalizable insights.

Overall, the authors are investigating a very important topic. However, several problems in experimental design as well as failure to address differences between their and previously reported results lessen the potential importance of this study.

We appreciate Reviewer #1's recognition that we have addressed an important topic, as well as the thorough evaluation of our manuscript's findings. These comments were insightful and ultimately helped us improve the manuscript significantly. We have addressed the reviewer's concerns by doing a better job of placing our results in context of the studies that the reviewer has mentioned, showing how this work is not in conflict, but complementary to what is already published in the field. In addition, we have done our best to provide strong evidence to support our positions that were not convincing to the reviewer. We hope that our modifications to the manuscript as well as well researched responses will remove any reservations and convince the Reviewer that the paper deserves publication.

1) GLP-1 incretin effect and insulin changes in microbiota-depleted mice do not agree with a study by Wichmann et al., 2013 (ref. 3) who extensively studied this relationship and concluded that the role of colonic GLP-1 was primarily on intestinal transit in the large intestine and not an incretin effect because the excursion of glucose was similar in both groups after exendin9. They showed SCFA decrease related to GLP-1 increase in microbiota-depleted mice evoking a feedback mechanism of slower gut transit. Neither intestinal transit was examined nor was the discrepancy addressed in the current manuscript. - Glp1r KO mice should be used to test the contribution GLP1 as has been done in ref. 3.

The reviewer makes several statements in this comment that we would like to address individually.

(a) "GLP-1 incretin effect and insulin changes in microbiota-depleted mice do not agree with a study by Wichmann et al., 2013 (ref. 3) who extensively studied this relationship..."

In Wichman's article, there are only a few panels investigating the effects of antibiotic depletion on mouse physiology: Fig 2G, 2I, and Fig 4D. The focus of the article was to investigate metabolism in germ-free (GF) mice, not AIMD mice. Fig 2G and 2I show that their antibiotic treatment led to a decrease in luminal SCFAs, an increase in proximal colon Gcg mRNA levels, respectively. We have confirmed that AIMD mice are similar to GF mice in our study in Fig 3A and Fig 4C, both of which contain more information than the Wichman, et al., paper (i.e. in Fig 3A, we have the breakdown of different SCFAs, and in Fig 4C, we have Pcsk7 expression, which

further suggests GLP-1 production). The differences in SCFAs and Gcg mRNA expression is far more pronounced in our AIMD mice, likely because we have attained microbiome depletion, which the Wichman paper does not seem to have achieved. Hence, we do not see a disagreement in the results that are directly comparable, and our results complement their findings.

To address this concern, we have made the following change to the introduction where we describe the Wichmann findings:

“In fact, antibiotic treated mice have decreased luminal SCFAs.”

(b) “...and concluded that the role of colonic GLP-1 was primarily on intestinal transit in the large intestine and not an incretin effect because the excursion of glucose was similar in both groups after exendin9. They showed SCFA decrease related to GLP-1 increase in microbiota-depleted mice evoking a feedback mechanism of slower gut transit.”

To clarify, Wichman, et al. did not show that GLP-1 is elevated in antibiotic treated or AIMD mice, only in germ-free mice. The Reviewer may be assuming that AIMD and GF are similar entities. Before our manuscript, there was little evidence that AIMD and GF had similar metabolic properties and, as explained in our introduction, there were paradoxes in the published literature that had led many to conclude that AIMD would cause opposing effects on host glucose homeostasis. Nevertheless, we acknowledge that, in the case of GLP-1 levels and insulin response, there are similarities between AIMD and GF mice, though we still do not see any disagreement with Wichman, et al.’s work. This may be because we do not agree with the Reviewer’s summary. Wichman, et al., based on the response of GTT to Ex9 in GF and conventional mice (their Fig 4A), conclude that “there are underlying differences in glucose metabolism in GF and [conventional mice], but the relative incretin effect of GLP-1 appears to be similar.” In other words, there is an increase in incretins in GF mice, but the response to incretins is similar between GF and conventional mice (i.e. the GF mice do not have a “stronger” response to GLP-1... they have a similar response). We acknowledge that we have not been as concise in our statement, but our observation is the same (see Fig 4I). Second, Wichman, et al. studied the effects of increased GLP-1 on GI motility and concluded, “... we found that GF mice exhibited significantly slower **small intestinal transit** in comparison to [conventional] controls.” (emphasis added). The authors do not measure, nor comment on intestinal transit in the large intestine as the Reviewer claims. Nevertheless, the effects of increased GLP-1 on GI motility is well known (PMID 11173902) and although this slowed motility is thought to influence appetite, the effects of GI motility on glucose homeostasis does not appear to overshadow the incretin effect. Wichman, et al. certainly make no conclusion or speculation about the relationship between SCFAs, GI motility and glucose homeostasis.

In response to the reviewer’s comments, we have made the following change to the results:

“In other words, the relative incretin effect of GLP-1 appears to be similar between AIMD and vehicle-treated mice, which suggests that lower blood glucose in AIMD mice is not solely caused by excessive GLP-1.”

(c) “Neither intestinal transit was examined nor was the discrepancy addressed in the current manuscript. - Glp1r KO mice should be used to test the contribution GLP1 as has been done in ref. 3.”

Wichman, et al. observed the slowed intestinal transit in GF mice was driven by increased GLP-1, but did not attribute any metabolic effect on this slowed transit. Since we are interested in glucose homeostasis in AIMD mice, and there is no direct relationship between GI motility, per se, and host glucose homeostasis, we believe characterizing intestinal transit in AIMD mice (and using Glp1r KO mice) is outside the scope of our paper and does not contribute to a better understanding of our central question: What is the effect of AIMD on glucose homeostasis?

2) Depending on antibiotic cocktail, the administration route and the baseline microbiota, the effects might be different in different protocols of AIMD and would not necessarily agree with the complete microbiota depletion (germfree). Although gammaproteobacteria in general would be expected to dominate, particular species might differ, which can result in different outcomes.

There may be some confusion about antibiotic treated mice and antibiotic induced microbiome depletion (AIMD), which is a separate entity. The distinction between the two is that AIMD protocols show evidence of microbiome depletion (e.g. negative cultures, decrease in stool DNA, engorgement of cecum), whereas most antibiotic protocols do not. We have used the most commonly used, and well-researched, protocol for AIMD (ref. 22, which has been cited more than 120 times). Reference 22 clearly shows that other antibiotic treatment protocols are ineffective in causing microbiome depletion and hence should not be considered AIMD. We have replicated these results in our lab; we performed experiments with once-daily dosing of quadruple antibiotics, and twice daily dosing of each one of four antibiotics alone. These regimens did not achieve microbiome depletion nor have a measurable effect on glucose homeostasis. Hence, we believe that the AIMD results are generalizable to any AIMD protocol where 3 or more broad spectrum antibiotics are used with the eradicating both Bacteroidetes and Firmicutes (e.g. see ref 3, 6, 18, 21).

To address the Reviewer's concern, we have made the following change in the introduction:

“AIMD is a distinct form of antibiotic treatment where through the administration of multiple broad-spectrum antibiotics/antifungals, the microbiome is depleted of all major bacterial phyla except Gammaproteobacteria, as evidenced by negative stool cultures, decrease in stool DNA, and engorgement of the cecum.”

3) increased glycolysis has been reported in the gut after microbiota introduction into germfree mice (PMID 22722618). This also seems to contradict the results of the current manuscript bringing up again the possibility of confounding effects of antibiotics or some other variables.

In the paper cited by Reviewer #1, the authors investigate the effects of conventionalization of a germ-free mouse on gene expression in the **jejunum**. The authors found that with conventionalization, the jejunum upregulated glycolysis transcriptional pathways. Implicit in the Reviewer's argument is that the jejunum and cecum/colon are similar organs. However, there is decades of research that show that these are clearly two different organs with different physiological and functional properties (see Comprehensive Physiology by APS, Supplement 19; <http://www.comprehensivephysiology.com/WileyCDA/Section/id-420623.html>) The most salient difference is that SCFAs are the preferred substrate for colonocytes, where butyrate is metabolized in preference to glucose. However, in the intestinal epithelial cells (e.g. in the jejunum), the primary substrate is glucose. Although we cannot comment on the effects of microbiome depletion (or reconstitution) on jejunal transcription, our findings that the upregulation of glycolysis and downregulation of fatty acid oxidation in colonocytes occurs in a setting of the luminal depletion of its preferred substrate, butyrate, does not contradict this excellent article, rather it complements it.

To address the Reviewer's concern, we have made the following change in the discussion:

“This may be a unique process to the colon, since in microbiome depleted states, the jejunum, which does not primarily use SCFAs for energy, increases glycolysis.”

4) the main claim of the gut as a ‘glucose sink’ is not sufficiently corroborated. It could be directly compared the measurements of tissue glucose levels between control, AIMD and germfree mice.

We agree with the reviewer that this is conjecture. We have removed this term from the abstract.

5) how can downregulation of gut gluconeogenesis suggested here be reconciled with studies showing that intestinal gluconeogenesis is important for improved systemic glucose tolerance (work by Mithieux lab and others)?

Mithieux lab have shown that **jejunal** gluconeogenesis is important for systemic glucose tolerance. We have not studied the jejunum as part of this manuscript and hence cannot comment on whether intestinal gluconeogenesis is affected by AIMD. As noted before, the jejunum and cecum/colon have different physiological characteristics and different metabolic machinery. Hence, we cannot assume that their response to AIMD will be similar.

6) direct effect of antibiotics on mouse cells cannot be excluded. For example, authors observed a discrepancy between AIMD and GF mice (figs. 4, S4). Can those be related to mitochondrial and other metabolic deficiencies intestinal epithelial or other cells as previously reported (ref.6)? Comparison to GF is quite limited in this study. In particular, in the discussion there is a statement that switch to anaerobic glycolysis cannot be attributed to mitochondrial deficit. Why? With the defect in mitochondrial oxidative phosphorylation, cells are trying to compensate energetic demands by boosting glycolysis.

The Reviewer brings up an excellent point. Morgun, et al. (ref. 6) found that both antibiotics and proteobacteria independently can negatively affect mitochondrial function. Hence, a defect in mitochondrial oxidative phosphorylation can contribute to the observed physiology, and we have revised the section as the Reviewer has recommended. Implicit in the Reviewer's comment is a desire for more comparisons between AIMD and GF mice physiologically. However, that is not the goal of our study. We aimed to assess the metabolic effects of AIMD to understand how it may affect metabolic experiments performed in these mice.

To address the Reviewer's concern, we have made the following change in the discussion:

“Previous studies which studied the effects of antibiotic administration in GF mice and transplantation of antibiotic-resistant bacteria in GF mice, show that both conditions decrease active mitochondria and inhibit mitochondrial gene expression, which could be contributing to increased epithelial cell death.⁶ We have observed a similar phenomenon where mitochondrial number and gene expression are decreased in AIMD mice. This could also be contributing to increased anaerobic glycolysis observed in AIMD mice.”

7) This work represents a rather extensive description of an AIMD model, which would be not feasible to implement for most labs (gavage 2 times a day every day for 30 days!). More importantly, it is not justified why such protocol is needed, considering enormous number of studies that have been using simpler protocols (drinking water) for more than a decade.

Please see the response to Reviewer #1, Comment (2). Ref 22 clearly shows that most protocols that are employed by researchers, including the ones that the reviewer suggests, do not achieve microbiome depletion. After publication of Ref 22, AIMD protocols have been standardized to avoid confusion and results that have not been replicated. As mentioned in comment (2), AIMD is distinct from antibiotic treatment. It is the responsibility of each investigator to show that their AIMD protocol actually achieves microbiome depletion. We have shown in Fig 1 and S1 that we have achieved microbiome depletion with our protocol. Our own work, and those published by others show that treatment with anything less than twice daily dosing of at least three broad spectrum antibiotics does not achieve microbiome depletion. Hence, studies that do not use these rigorous antibiotic regimens should be interpreted carefully since, although they have decreased microbiome diversity, they have not attained microbiome depletion per se. Finally, a gavage of 30 days is not necessary to observe the metabolic effects noted. In fact, experimental replicates for shorter duration (i.e. 13 days) found similar metabolic effects. Our methods section has now been updated to reflect this point.

8) regarding the microbiome analysis (fig. 1): why showing together Bacteroidetes & Firmicutes? Which genera/species of proteobacteria were detected in AIMD mice?

The goal of AIMD protocol is to deplete the major phyla of the microbiome (i.e. Firmicutes and Bacteroidetes). Fig 1B and 1C show that in the most concise manner. If the reviewer is interested

in specific phyla levels, that is already depicted in Fig 1D and Fig S1F. The particular proteobacteria that survive the AIMD protocol are listed in Table S1.

9) What if corrected by decreased overall absolute abundance, would amount of proteobacteria differ between groups, i.e. absolute amount per g of stool?

This is an interesting question and one that we are curious about as well. We can only speculate based on the available data. Fig S1B shows that cecal DNA decreased 20-fold between AIMD and Vehicle-treated mice. In contrast, we see the proportion of proteobacteria increase 10,000-fold. If 16S is proportionally the same in the cecal DNA extracted from both groups (which is unlikely to be a correct assumption), it is likely that proteobacteria are flourishing in the microbiome depleted environment perhaps due to a lack of competition. However, since this is purely speculative, we have not added this information to the manuscript.

10) cecal weight (and likely other parts of distal gut) is increased in almost 2 g in AIMD mice but body weight did not differ from controls. Something is missing. Decreased leptin levels (fig4) indicate that there might be less fat.

This concern was shared with Reviewer #3. We are confident of the body composition data that we have reported for reasons. First, we would like to point out that the body composition data has been replicated (Fig 2H and Fig S2C & D). To address this important concern, we took a closer look at our body composition data. We noted that although the weight of the mice during the experiment was not significantly different (Fig 2F and S2A), on the day of body composition assessment the weight of the vehicle-treated mice was non-significantly less than that of the AIMD mice. We attribute this difference to the fact that weekly weights were done while the animal had ad libitum access to food, but on the day of body composition testing, the mice were fasted. Upon dissection of the mice, we noted that the ceca of AIMD mice were full of watery stool but did not note any difference in subcutaneous or visceral fat. When we reanalyzed the body composition data (please see new Fig 2H), we noted that the non-fat/non-lean mass compartment was significantly different between the two mouse groups. We attribute this difference to differences in cecal content.

Without performing more detailed analysis of the white adipose tissue, it will be difficult to know what is causing the decrease in circulating leptin. We speculate that there are two reasons for the decreased serum leptin. First, besides adiposity, leptin is released in a fed state (i.e. high circulating glucose) and increased serum insulin. Although fasting insulin is similar between AIMD and vehicle-treated mice, there is a significant drop between fasting and fed insulin in AIMD mice, who are also hypoglycemic, both of which could explain decreased leptin levels in AIMD mice. Another potential reason is the high levels of GLP-1 may inhibit leptin release. Leptin is known to be a potent stimulator of GLP-1 secretion from L-cells (PMID 12540594; 25833771). It is reasonable to speculate that since high GLP-1 is a potent appetite suppressor, decrease leptin levels help regulate food/energy intake.

To address this comment, we have split our body composition graphs into three separate ones showing differences between lean, fat, and non-fat/non-lean mass. In addition, we have modified the discussion to include:

“Furthermore, AIMD induces decreased fasting serum leptin levels without a change in body adiposity. Besides body adiposity, a fed state (as determined with high circulating glucose) and increased serum insulin can both increase leptin release. Although fasting insulin is similar between AIMD and vehicle-treated mice, there is a significantly more precipitous drop between fasting and fed insulin in AIMD mice who are also hypoglycemic, both of which could explain decreased leptin in these mice. In addition, GLP-1 is known to affect satiety and high levels decrease food intake as we have observed in some of our cohorts. High potential levels of GLP-1 may inhibit leptin release. Leptin is known to be a potent stimulator of GLP-1 secretion from L-cells.^{29, 30} Because high levels of serum GLP-1 is a potent appetite suppressor in our AIMD

mice, decrease in serum leptin levels may be a response to help regulate food/energy intake. Hence, the relationship between GLP-1 on adipose tissue in general, and leptin release in particular should be more closely investigated.”

11) regarding fig. 3: not all primary BA were lower, some were higher. All abbreviations of figs. 3B,C need to be explained. fig 3C is not the best representation. Why make fig. S3? It can be combined with main fig 3.

We agree with the reviewer that our statement was confusing and have corrected it. The abbreviations for all the bile acids are now in the figure legend. We have removed Fig S3 as requested by Reviewer 2.

12) regarding transcriptomic data: it would be important to know if AIMD transcriptomes are very different from GF or not in order to address the concern of the effects of antibiotics on mammalian tissues. Also, whether immune alterations can be attributed to antibiotic-resistant microbe outgrowth.

The goal of our study was to determine the metabolic effects of AIMD, particularly on glucose homeostasis. Hence, a comparison between AIMD and GF mice is outside the scope of this article. We would like to point out that there is already a very well-written article on gene expression differences between AIMD and GF mice (ref 6), hence performing these experiments would not be novel nor help us attain a better understanding of AIMD effects on metabolism.

13) levels of fasting insulin might be actually different in fig S4a at fasted state with means ~1.5 vs ~0.5. Alas, this was not significant with the p-value for this comparison was 0.12.

14) was the effect specific to TCA? TbmCA was also increased in AMD mice.

We chose TCA since it was remarkably different between AIMD and Vehicle treated mice (7.4-fold difference). The effect of TbmCA was far smaller (3.6-fold difference). Furthermore, the effects of TbmCA on host metabolism have been investigated (Ref. 5) whereas the effects of TCA are somewhat unexpected. See also response to Reviewer #3, Comment #7. We have been inspired by these results to investigate the effects of bile acids on glucose homeostasis much more closely. Hence, it is outside of the scope of this study and we will have more information on the matter in the future.

15) Would the phenotype be the same after oral GTT?

We have shown an OGTT in Fig 4I. AIMD mice still have a lower glycemic response in a OGTT compared to vehicle treated mice.

16) several lipid metabolism genes show increased expression in AIMD mice (fig.6a), e.g ApoB, ApoA4, Hsd17b12 etc. What's the interpretation of this?

It is difficult to know why these genes are upregulated without further exhaustive experiments that are outside the scope of this manuscript. The reason why some of the genes are upregulated is likely due to multifactorial reasons. For example, ApoB, a protein responsible for shuttling fat, may be upregulated as a compensatory mechanism since the colonocyte is searching for more fat. ApoA4 has antioxidant effects that likely help the colonocyte with damaged mitochondria. We do not wish to include these speculations in our manuscript since we do not have evidence to support them and they do not contribute to the central aim of the paper.

17) using t-test might not be appropriate if the data did not have normal distributions, which might the case at several instances.

We assumed that circulating hormones/cytokines are normally distributed which is supported by a multitude of previous studies. Comparing the data using non-parametric comparisons (i.e. Mann Whitney U test) did not change any of the conclusions and in most cases further reduced the p-

values. Certain measures for which we were unsure (e.g. bile acid pool, short-chain fatty acids) we used non-parametric comparisons (e.g. Figure 3).

18) abstract needs to mention analyses of microbiota

We have included information on the effects of AIMD on the composition of the microbiome.

19) it is not stated how many times the experiments were repeated and although stated, data are not shown for different days of duration of the protocol giving "identical" results.

We now include in our methods information about the number of experiments, the length of each experiment and which samples were used for each study. The experiments were repeated four times with twice-a-day dosing, and once with once-a-day dosing. The latter experiment did not achieve microbiome depletion and those results were excluded. The remaining four times were done at the following lengths: 30 days, 19 days, and 13 days (twice). We have mentioned lengths of the experiment for most figures (e.g. Fig 2), but will include more details in the methods due to the reviewer's request.

Reviewer #2 (Remarks to the Author):

In this manuscript, the authors describe the effect of depleting the microbiome using antibiotics on metabolic homeostasis. This is an important topic, regarding the numerous studies demonstrating a contribution of the gut microbiota in metabolic diseases as well as the increasing number of projects involving gut microbiota transplants. Using a mice model, they show that Antibiotic induced microbiome depletion (AIMD) decreases baseline serum glucose levels, improves glucose tolerance and insulin sensitivity without altering food intake or body adiposity. Additional analyses including assays of short chain fatty acids, bile acids and gut hormones as well as cecal transcriptome analysis lead to the hypothesis that AIMD shifts the gut metabolism toward glucose utilization as an explanation of the observed phenotype.

The findings are interesting and certainly very well described. Overall, the paper is well written and the results represent significant additions to previously-published studies. Nevertheless, several important concerns can be inferred:

We appreciate Reviewer #2's recognition of the importance of the study, and appreciation of our hard work. We appreciate the Reviewers thorough evaluation of our manuscript and the comments have helped improve the manuscript tremendously. We have addressed the reviewer's concerns by adding the additional details requested. We hope that our responses will remove any reservations and convince the Reviewer that the paper deserves publication.

Major comments:

1) The stool cultures presented in the paper (Fig 1A and S1a) have not been done properly. Bacterial population in mouse feces varies between 10^{11} to 10^{12} bacteria/g feces. Even if a part of the microbiota is not culturable, a minimum of 10^{10} (10 billions) cfu/g of feces is needed to have a representative idea of the bacterial population. Were the cultures done anaerobically? If not, this is completely useless as the great majority of the gut bacteria are strict anaerobic microbes. Moreover, LB is definitely not a medium for culturing gut bacteria. Therefore, the results presented in Fig 1A and S1A does not tell anything about the bacterial population in the mice gut and must be deleted. Proper stool cultures must be done anaerobically using different culture media and lead to at least 10^{10} cfu/g feces.

We are well aware that the LB culture does not grow the vast majority of luminal organisms, most of which are anaerobic. The purpose of depicting the stool cultures is to show microbiome depletion, not to culture any specific organism from the stool. That is, culturing stool on LB agar yields colonies of bacteria that disappear with AIMD. 16S characterization is a better test to show depletion of specific organisms. This is a test used by others who have shown microbiome

depletion (e.g. ref 22). We have included it for thoroughness. However, if this is potentially confusing to the reviewers/readers, we can certainly exclude the figures.

To address the Reviewer's concern, we have modified the following sentence in the Methods:

“To assess microbiome depletion, fresh fecal samples were collected in pre-weighed tubes containing 1mL PBS and their weights measured.”

2) Similarly, the authors assayed the concentration of DNA/g of stool. This is not clear why they did not measure bacterial population using real-time PCR using this DNA. This would tell which percentage of the bacteria has been depleted by AIMD.

We agree with the reviewer that assessing 16S DNA would have given more insight into stool bacterial content. Whereas performing this type of 16S quantification is helpful in mice that are not microbiome depleted, it proved to be a challenge in AIMD mice. The low levels of DNA in the AIMD feces reached the limit of accurate quantification with Nano-Drop, and we did not feel confident in our measures of 16S at that low level. Although DNA/g of stool is more crude measure, it clearly shows the main effect, that our antibiotic cocktail attains microbiome depletion. Furthermore, it confirms that all microorganisms (including fungal species) have been depleted.

3) Experimental procedures indicate that AIMD protocol was maintained for different length of time (13 to 30 days) leading to identical results. However, it is not clear which duration was used for the represented results.

See response to Reviewer 1, comment #19

4) Oral gavage every 12 hours must create a stress in the mice. It would have been interesting to measure the parameters that can be assayed on living animals (glycemia, insulinemia, glucose tolerance, fecal metabolites...) before the start of AIMD to evaluate effects of gavaging, independently of the effect of the antibiotics.

Although we agree with the reviewer that it would be interesting to measure the effects of gavage on glucose homeostasis independently, it is outside of the scope of this study. It is important to point out that our control mice with whom we are comparing our AIMD mice underwent gavage twice daily as well, hence making antibiotic exposure the only variable that was different between the two groups. In addition, our depletion results have replicated those of past researchers who have established this protocol (Reikvam, et al. Plos One 2011) as well as those who have used this protocol in other published studies. The best published evidence shows that mice who undergo awake daily gavage are not particularly susceptible to physiological endpoints of stress (adrenal gland weight, neutrophil:lymphocyte ratio, plasma corticosterone) compared to mice that received gavage under anesthesia (Jones, et al. J. Am. Assoc. lab Anim. Sci 2016 - PMID 27931321).

5) Gene expression of gut hormones have been analyzed in cecum. While cecal transcriptome analysis is globally relevant, the main expression site of some of these hormones is not cecum. Therefore, it would be more relevant to analyse expression of ghrelin in the stomach, and of CCK and GIP in the duodenum.

The only gene expression of a gut hormone we have included in the manuscript is that of Gcg in cecum, and Pyy and Cck to show that genes of hormones that are co-released with GLP-1 are also highly expressed. All other gut hormones shown were actual serum levels of the gut hormones, not the gene expression of gut hormones in the cecum. We have further clarified this in our methods and results. However, we can exclude the Pyy and Cck gene expression if this is confusing.

6) The methods section about the microbial community analysis is uncomplete. Specifically the number of raw sequences, the number of chimeras, filtered sequences due to other quality reasons, rarefaction of

data, number of minimum sequences from each sample etc. should be spelled out as part of supplementary material. The number and description of OTUs shared by Pre-treatment vs AIMD vs vehicle must be specified. Also Beta diversity should be indicated.

We have added the number of raw and filtered sequences, and rarefaction of data to the methods as requested. This data shows that we have significant depth in our sequencing to do the analyses in our manuscript. Because we did close-reference OTU clustering, we do not have chimeras. In addition, we have added a figure which includes more information about α - and β -diversity based on UniFrac distances. In addition, below we have added a Venn Diagram of shared OTUs. We have not included the description of OTUs differences since we are limited by number of words and figures and we do not think it is relevant to the physiological effects we are observing. Nevertheless, this information is easily available by reviewing the supplemental tables we have provided with the manuscript.

To address the Reviewer's comments, we have added a new Fig S1G which has a Venn Diagram of shared OTUs and S1J which now includes within- and between-group UniFrac distances. In addition, we have modified the methods to include the information the Reviewer requested.

7) It has been reported that "metabolic endotoxemia", defined as a moderate increase in the concentration of bacterial lipopolysaccharide (LPS) in the plasma, is a triggering factor linking the gut microbiota to metabolic disorders. Plasma LPS concentration should be assayed to decipher if the effect of AIMD on glucose homeostasis is due, at least in part, to reduced endotoxemia. Similarly, altered gut barrier has been implicated in the gut bacteria induced metabolic disorders. Gut permeability analysis as well as expression of tight junction proteins would be useful to determine if AIMD influence gut barrier function.

The reviewer brings up an interesting point about the effects of leaky gut and circulating bacterial proteins on metabolism. However, the effect of AIMD on serum LPS levels have been documented (ref 18). In addition, our transcriptomic data shows that genes for tight junction proteins are highly upregulated (see Tables below). We acknowledge that glucose homeostasis is affected by many different ongoing biological processes. Given space constraints imposed by the journal, we believe that including additional data on LPS/endotoxemia contribution to glucose homeostasis is too unrelated to the other experiments/studies we have included and hence will be lost in the data and not receive the attention it deserves, especially in context of work that has been already done on AIMD and LPS.

To address the Reviewer comment, we have added the following to discussion:

"...and lower serum LPS,¹⁸ which could affect glucose homeostasis through decreased endotoxemia.²⁶"

Occludin
Ocln

Gene	Mean normalized count	Fold change (log2)	Raw p-value	Adjusted p-value
Ocln	1323.77	1.33	7.20E-10	8.04E-08

Claudin
Cldn1...25

Gene	Mean normalized count	Fold change (log2)	Raw p-value	Adjusted p-value
Cldn7	6510.57	0.47	1.24E-02	6.78E-02
Cldn15	2541.91	-0.80	1.82E-06	7.07E-05
Cldn23	932.50	0.18	4.25E-01	6.40E-01
Cldn12	907.54	0.15	4.30E-01	6.44E-01
Cldn3	825.52	0.47	1.67E-01	3.71E-01

Cldnd1	625.86	0.72	5.73E-04	6.92E-03
Cldn4	595.67	0.47	2.32E-01	4.54E-01
Cldn2	326.04	-1.34	6.56E-05	1.30E-03
Cldn8	95.74	1.55	2.21E-05	5.40E-04
Cldn5	13.19	-0.69	1.13E-01	2.93E-01
Cldn10	10.71	-0.51	4.27E-01	6.42E-01
Cldn1	9.13	0.05	9.49E-01	9.76E-01
Cldn13	6.19	-2.28	1.05E-02	6.00E-02
Cldn22	1.16	-2.50	1.36E-01	
Cldn20	0.92	-5.92	4.64E-02	
Cldn14	0.75	-1.48	3.88E-01	
Cldn11	0.69	-2.09	3.37E-01	
Cldn34c1	0.60	0.40	8.29E-01	
Cldn34b2	0.29	-0.59	8.45E-01	
Cldnd2	0.11	2.33	6.46E-01	
Cldn9	0.10	-4.07	4.30E-01	

JAM
Jam2
Jam3

Gene	Mean normalized count	Fold change (log2)	Raw p-value	Adjusted p-value
Jam3	48.51	-0.44	2.45E-01	4.69E-01
Jam2	45.59	0.25	5.84E-01	7.65E-01

F11r (Jama)

F11r	4040.27	0.11	5.37E-01	7.31E-01
------	---------	------	----------	----------

Tricellulin
Marveld2

Marveld2	275.61	0.26	1.21E-01	3.05E-01
----------	--------	------	----------	----------

8) Several information on animal experiment procedures are lacking: How were sacrificed the animals ? How many mice per cage...? Also information showing that guidelines for the care and use of laboratory animals were respected and approved by an ethical committee must be provided.

We have added additional information about the method of euthanasia and mouse housing. In addition, we have been explicit in our methods section that the care and use of laboratory animals was reviewed by the Salk Institute IACUC committee.

Minor comments:

9) line 70: in humans too !.

We have removed the qualifier “In mice”

10) line 337: Serum BA are expressed nmol/mg ? mg of what ? or is it ml ?

Thank you for pointing out this error. The proper concentration is μM which we have now included in the manuscript.

11) line 397: reference must be provided

We have included the reference.

12) line 722: Bacteroidetes instead of Bacteroidetes (also in Figure 1 legend).

We have made these corrections.

13) Line 421: It is stated “AIMD studies can help find an oral agonist of incretin signaling”. It is not clear what the authors mean.

We have clarified the sentence.

14) Table S3 shows that AIMD increases some of the primary BA (TCA, TmMCA) and decreases some other (CA, aMCA, TaMCA). This must be discussed.

Please see our response to Reviewer #1, Comment 11.

15) The authors compare some of their results to what has been obtained in GF animals. GF animals are widely used for gut microbiota transplants experiments leading to the demonstration of a causative relationship between the gut microbiota and host phenotype. Do the authors believe that AIMD is a better model than GF for microbiota transplant? This must be argued and discussed.

The reviewer raises an interesting question that we have considered as well. However, our results do not show whether AIMD over GF mice are more appropriate for metabolic experiments involving microbiota transplants and hence we are reluctant to speculate. Our results simply point out that, like GF mice, AIMD mice have a highly abnormal metabolic background that must be considered when they are used for metabolic studies.

16) Isovaleric and caproic acids are increased in AIMD. Please comment and propose an explanation.

The reviewer brings up an observation in SCFAs that also piqued our interest. Although we suspect that higher levels of these SCFAs is the result of incomplete fermentation to butyrate, we do not have enough evidence in our results to make an informed statement. Since this is speculative, we have not included it in the manuscript.

17) AIMD mice have less blood leptin while similar fat mass than vehicle-treated mice. This must be discussed.

Please see our response to Reviewer #1, Comment 10.

18) Globally the effects of specific BAs are poorly described. Also no information is given on TGR5 expression and its potential involvement in the observed effects. Please see some reviews (Chávez-Talavera O et al, Bile Acid Control of Metabolism and Inflammation in Obesity, Type 2 Diabetes, Dyslipidemia, and Nonalcoholic Fatty Liver Disease, *Gastroenterology*. 2017,152(7):1679-1694 ; Gérard P. Metabolism of Cholesterol and Bile Acids by the Gut Microbiota. *Pathogens*. 2014; 3(1):14-24....) and comment your results accordingly.

We agree that bile acids are likely affecting host metabolism through TGR5 though these mechanisms are still being explored by us and others.

In response to the Reviewer's comments, we have made the following changes to the discussion: “Although TGR5 gene expression levels were not different in the cecum of AIMD- and vehicle-treated mice, it is highly likely that AIMD-induced BA changes are also affecting TGR5 signaling which could also influence glucose homeostasis.^{32, 33}”

19) In vehicle treated mice, Bacteroidetes increases and Firmicutes decreases compared to pre-treatment. Is it significant? Effect of stress due to gavage? Please comment.

This difference is not significant. Four mice from the pre-treatment group had an increase in Firmicutes which could be due to a cage effect. We do not believe the pre-treatment cage effect is

relevant to our results since the post-treatment samples, where we did all of our physiological studies, show no cage effect.

20) Fig S1F: Two different groups seem to appear regarding composition of the microbiota in Pre-treatment mice. It is important to consider possible "cage effects" in rodent studies, especially when examining the microbiome. It is known that microbiota of cage mates are correlated, and in some cases, cage effects have been shown to exceed the effects of exposures under study. One point the authors do not address is the housing of mice, and potential housing effects. Please indicate how the mice were housed during the normal course of the experiment and if the differences seen in this figure is related to a cage effect.

Please see our response to Reviewer #2, Comment 8 and 19. We agree that there is a cage effect between the two cages we used in the pre-treatment condition. However, as is evidence in Fig 1E, S1F, S1J, the cage effect in this case did not exceed the effects of the exposure, mainly because the effects of AIMD are so pronounced.

21) Fig 2 legend: cumulative instead of cunultative

Thank you for pointing out this error. We have corrected it.

22) FigS3 is not informative. It may let think that primary BA increased and secondary BA decreased in AIMD while the ratio is very in AIMD because secondary BA are extremely low.

We have removed Figure S3.

23) Fig4 and S4: controls in blue must be defined

We have added a description of controls in our methods and in the text.

24) Fig 7: TUDCA is increased in AIMD serum. This should be highlighted and discussed, especially because this BA is relevant in term of health and disease and may explain some of the effects of the gut bacteria on host metabolism (see Llopis et al, Gut, 2016, 65(5):830-9 as an example)

We agree with the author that this is interesting. To address the Reviewer's comments, we have added the following sentence to the discussion"

"Interestingly, AIMD also increases TUDCA, a BA with known hepatoprotective properties that could also play a role in decreasing inflammation.³⁴ Hence, AIMD-induced changes in BA physiology could likely affect host metabolism through various methods, most of which remain to be explored."

Reviewer #3 (Remarks to the Author):

In this manuscript, Dr. Zarrinpar and his/her colleague reported that antibiotic induced microbiome depletion alters metabolic homeostasis by affecting gut signaling and colonic metabolism. This research showed that AIMD improves insulin sensitivity without altering food intake or body adiposity through decrease of butyrate and secondary BA.

We appreciate Reviewer #3's thorough evaluation of our manuscript and the comments have helped improve the manuscript tremendously. The Reviewer brings up many of the same concerns as previous Reviewers which we have addressed. The Reviewer main concern is whether our body adiposity measures are correct. If it is not, the Reviewer worries that all our measures could be explained by this single mis-measure. We have used an EchoMRI to assess body composition used by the vast majority of metabolic phenotyping cores as well as metabolic labs. We have replicated our own body composition results and have presented both in the manuscript. In addition, a change in body adiposity alone would not explain the vast majority of findings we have in our manuscript. We hope that our responses will remove any reservations and convince the Reviewer that the paper deserves publication.

1) This experimental data and the interpretation are insufficient and the results do not present logical representation. They insist that AIMD did not alter food intake (Fig2G) and body adiposity (Fig2F, H, FigS2ACD). However, Food intake of AIMD tends to be low compared with vehicle.

After acclimation to gavage, the food consumption in AIMD mice was the same as vehicle-treated mice (Fig 2G and S2B). Hence, we have presented replicated results showing food intake does not differ in AIMD mice. We have also shown in replicated results that body weight does not differ between AIMD and vehicle-treated mice (Fig 2F and S2A). As we stated in our response to Reviewer #1 Comment (10), the body composition results showing no difference in lean and fat mass was replicated in our lab in two different cohorts was the same between AIMD- and vehicle-treated mice (Fig 2G, S2C, S2D). We hope the fact that we have replicated our own results in two different cohorts, with a total of more than 18 mice per condition across both replicates, using conventional techniques and equipment, will remove the Reviewer's skepticism of these results.

2) Moreover, although the cecum weight of AIMD is more than 2g (FigS1D), both body weights are same. This is discrepancy. Therefore, their AIMD phenotype can be explained by the difference of adiposity.

Please see response to Reviewer #1, comment (10) regarding the body composition and leptin release. In short, most of the cecal weight is likely in the non-fat/non-lean mass compartment, which we found to be different between our vehicle- and AIMD-treated mice. However, we would like to point out that, even if our mice had decreased adiposity (**which they don't**), it would by itself not explain why AIMD mice have hypoglycemia, increased insulin sensitivity, fecal and serum bile acids, fecal short-chain fatty acids, increased incretin release, bile acid signaling or any of our transcriptomic results. Hence, the bulk of the evidence points to an altered glucose homeostatic change due to altered gut signaling and the main findings of our manuscript are still valid regardless of body adiposity.

3) I think that GLP-1 high concentration despite SCFAs and BA low concentration in AIMD is also due to this reason.

Although there is a volume of literature investigating the effects of GLP-1 on white adipose tissue, literature search in Pubmed and Google Scholar did not reveal any mechanism by which white adipose tissue can affect incretin release from the gut. If such a mechanism exists, we would appreciate appropriate references.

Other

4) Fig3A- What bacteria related on butyrate decrease by AIMD?

By analyzing our PICRUST data more closely, we can see which OTUs contribute to butyrate metabolism, are present in high numbers in our vehicle-treated and pre-treatment mice, but are absent from the AIMD mice. Since AIMD caused significant microbiome depletion, there were many hits in this analysis, encompassing most taxonomies. The biggest hit in this analysis were in the order Clostridiales, particularly in the family Peptostreptococcaceae.

5) Fig4 and Fig4S- Why does value of all hormones differ between Vehicle and Controls?

Control mice are conventional brethren of the GF mice used for the experiment. That is, they are raised in a different facility, different age, and different normal chow diet than vehicle-treated mice. In contrast to vehicle-treated mice, the control mice are also not gavaged twice daily. Please also see Reviewer #2, Comment 23.

6) Fig5- They should perform cecum histological analysis and cell assay as well.

We have performed cecum histological analysis and cell assays to understanding GLP-1 release in setting of AIMD. We could not find any quantifiable histological differences between AIMD and vehicle-treated mice and the cell assay work did not help identify specific metabolites/luminal substrates that could mimic the effect we have observed *in vivo*. It is unclear to us what type of cell assay work the reviewer has in mind. Since the intent of our study is to investigate the effects of AIMD on host metabolism, it is not clear any cell assay will help with our hypothesis. We acknowledge that new hypotheses are generated by our work (e.g. mechanism for GLP-1 release), and those should be explored further in different studies since they are outside the scope of this study.

7) Fig7E- They should examine aMCA, bMCA (Vehicle: high) and TbMCA (AIMD: high) in addition to TCA (AIMD: high).

The reviewer brings up an interesting point in which we are also interested in, that different bile acids can influence incretin release and glucose homeostasis. Although we agree that this is an important question, it is outside the scope of this paper which is focused on the effects of AIMD on glucose homeostasis. We are pursuing this question in our subsequent studies.

RESPONSE TO REVIEWERS' COMMENTS

Reviewer #1 (Remarks to the Author):

The work by Zarrinpar et al. characterized metabolic alterations evoked by a cocktail of antibiotics that results in dramatic reduction of gut microbiota in mice. The rationale is that a better characterization of this model is needed so that it can be used more in studies addressing the role of microbiota. Examining cecum, the authors came to the conclusion that there is higher glucose utilization in this part of the gut after antibiotics as a result of deficit in short chain fatty acids. Importantly, some of these results are not entirely novel (e.g. ref 3, 6) while others do not agree with previous studies (ref. 3, PMID 22722618, PMID: 24412651). Importantly, it is not clear which of the described effects are truly microbiota-dependent and which ones are context-dependent, i.e specific antibiotic mix, the baseline microbiota and the resulting antibiotic-resistant microbiota. Thus the effects might not be generalizable. Further, comparison to GF is very limited, making it hard to conclude which effects are related to microbiota absence vs. antibiotic toxicity vs. antibiotic-resistant microbes. In addition, the central claims regarding GLP-1, insulin are at odds with previous studies (ref. 3) and this fact has not been addressed.

Of note, the employed model of administration of Abx is very inconvenient and different from hundreds of other studies employing antibiotic-treated mice as a surrogate of germfree. Thus, it is doubtful that the results of this study that differ from other reports will provide generalizable insights.

Overall, the authors are investigating a very important topic. However, several problems in experimental design as well as failure to address differences between their and previously reported results lessen the potential importance of this study.

We appreciate Reviewer #1's recognition that we have addressed an important topic, as well as the thorough evaluation of our manuscript's findings. These comments were insightful and ultimately helped us improve the manuscript significantly. We have addressed the reviewer's concerns by doing a better job of placing our results in context of the studies that the reviewer has mentioned, showing how this work is not in conflict, but complementary to what is already published in the field. In addition, we have done our best to provide strong evidence to support our positions that were not convincing to the reviewer. We hope that our modifications to the manuscript as well as well researched responses will remove any reservations and convince the Reviewer that the paper deserves publication.

1) GLP-1 incretin effect and insulin changes in microbiota-depleted mice do not agree with a study by Wichmann et al., 2013 (ref. 3) who extensively studied this relationship and concluded that the role of colonic GLP-1 was primarily on intestinal transit in the large intestine and not an incretin effect because the excursion of glucose was similar in both groups after exendin9. They showed SCFA decrease related to GLP-1 increase in microbiota-depleted mice evoking a feedback mechanism of slower gut transit. Neither intestinal transit was examined nor was the discrepancy addressed in the current manuscript. - Glp1r KO mice should be used to test the contribution GLP1 as has been done in ref. 3.

The reviewer makes several statements in this comment that we would like to address individually.

(a) "GLP-1 incretin effect and insulin changes in microbiota-depleted mice do not agree with a study by Wichmann et al., 2013 (ref. 3) who extensively studied this relationship..."

In Wichman's article, there are only a few panels investigating the effects of antibiotic depletion on mouse physiology: Fig 2G, 2I, and Fig 4D. The focus of the article was to investigate metabolism in germ-free (GF) mice, not AIMD mice. Fig 2G and 2I show that their antibiotic treatment led to a decrease in luminal SCFAs, an increase in proximal colon Gcg mRNA levels, respectively. We have confirmed that AIMD mice are similar to GF mice in our study in Fig 3A and Fig 4C, both of which contain more information than the Wichman, et al., paper (i.e. in Fig 3A, we have the breakdown of different SCFAs, and in Fig 4C, we have Pcsk7 expression, which

further suggests GLP-1 production). The differences in SCFAs and Gcg mRNA expression is far more pronounced in our AIMD mice, likely because we have attained microbiome depletion, which the Wichman paper does not seem to have achieved. Hence, we do not see a disagreement in the results that are directly comparable, and our results complement their findings.

To address this concern, we have made the following change to the introduction where we describe the Wichmann findings:

“In fact, antibiotic treated mice have decreased luminal SCFAs.”

Agree but not sure why Fig. 3A is mentioned since it doesn't show germfree mice, only 4C does.

(b) “...and concluded that the role of colonic GLP-1 was primarily on intestinal transit in the large intestine and not an incretin effect because the excursion of glucose was similar in both groups after exendin9. They showed SCFA decrease related to GLP-1 increase in microbiota-depleted mice evoking a feedback mechanism of slower gut transit.”

To clarify, Wichman, et al. did not show that GLP-1 is elevated in antibiotic treated or AIMD mice, only in germ-free mice. The Reviewer may be assuming that AIMD and GF are similar entities. Before our manuscript, there was little evidence that AIMD and GF had similar metabolic properties and, as explained in our introduction, there were paradoxes in the published literature that had led many to conclude that AIMD would cause opposing effects on host glucose homeostasis. Nevertheless, we acknowledge that, in the case of GLP-1 levels and insulin response, there are similarities between AIMD and GF mice, though we still do not see any disagreement with Wichman, et al.'s work. This may be because we do not agree with the Reviewer's summary. Wichman, et al., based on the response of GTT to Ex9 in GF and conventional mice (their Fig 4A), conclude that “there are underlying differences in glucose metabolism in GF and [conventional] mice, but the relative incretin effect of GLP-1 appears to be similar.” In other words, there is an increase in incretins in GF mice, but the response to incretins is similar between GF and conventional mice (i.e. the GF mice do not have a “stronger” response to GLP-1... they have a similar response). We acknowledge that we have not been as concise in our statement, but our observation is the same (see Fig 4I). Second, Wichman, et al. studied the effects of increased GLP-1 on GI motility and concluded, “... we found that GF mice exhibited significantly slower **small intestinal transit** in comparison to [conventional] controls.” (emphasis added). The authors do not measure, nor comment on intestinal transit in the large intestine as the Reviewer claims. Nevertheless, the effects of increased GLP-1 on GI motility is well known (PMID 11173902) and although this slowed motility is thought to influence appetite, the effects of GI motility on glucose homeostasis does not appear to overshadow the incretin effect. Wichman, et al. certainly make no conclusion or speculation about the relationship between SCFAs, GI motility and glucose homeostasis.

In response to the reviewer's comments, we have made the following change to the results:

“In other words, the relative incretin effect of GLP-1 appears to be similar between AIMD and vehicle-treated mice, which suggests that lower blood glucose in AIMD mice is not solely caused by excessive GLP-1.”

(c) “Neither intestinal transit was examined nor was the discrepancy addressed in the current manuscript. - Glp1r KO mice should be used to test the contribution GLP1 as has been done in ref. 3.”

Wichman, et al. observed the slowed intestinal transit in GF mice was driven by increased GLP-1, but did not attribute any metabolic effect on this slowed transit. Since we are interested in glucose homeostasis in AIMD mice, and there is no direct relationship between GI motility, per se, and host glucose homeostasis, we believe characterizing intestinal transit in AIMD mice (and using Glp1r KO mice) is outside the scope of our paper and does not contribute to a better understanding of our central question: What is the effect of AIMD on glucose homeostasis?

Ok but please add that other factors like lower gut motility might be contributing.

2) Depending on antibiotic cocktail, the administration route and the baseline microbiota, the effects might be different in different protocols of AIMD and would not necessarily agree with the complete microbiota depletion (germfree). Although gammaproteobacteria in general would be expected to dominate, particular species might differ, which can result in different outcomes.

There may be some confusion about antibiotic treated mice and antibiotic induced microbiome depletion (AIMD), which is a separate entity. The distinction between the two is that AIMD protocols show evidence of microbiome depletion (e.g. negative cultures, decrease in stool DNA, engorgement of cecum), whereas most antibiotic protocols do not. We have used the most commonly used, and well-researched, protocol for AIMD (ref. 22, which has been cited more than 120 times). Reference 22 clearly shows that other antibiotic treatment protocols are ineffective in causing microbiome depletion and hence should not be considered AIMD. We have replicated these results in our lab; we performed experiments with once-daily dosing of quadruple antibiotics, and twice daily dosing of each one of four antibiotics alone. These regimens did not achieve microbiome depletion nor have a measurable effect on glucose homeostasis. Hence, we believe that the AIMD results are generalizable to any AIMD protocol where 3 or more broad spectrum antibiotics are used with the eradicating both Bacteroidetes and Firmicutes (e.g. see ref 3, 6, 18, 21).

To address the Reviewer's concern, we have made the following change in the introduction: "AIMD is a distinct form of antibiotic treatment where through the administration of multiple broad-spectrum antibiotics/antifungals, the microbiome is depleted of all major bacterial phyla except Gammaproteobacteria, as evidenced by negative stool cultures, decrease in stool DNA, and engorgement of the cecum."

I still do not agree with the term antibiotic induced microbiome depletion (AIMD), which gives an impression of total elimination of microbiota. But in fact, it is rather an *antibiotic-induced microbiome reduction* (AIMR). In fact, it is not true what authors responded: "evidence of microbiome depletion (e.g. negative cultures....., whereas most antibiotic protocols do not" .

First, it's clear from Fig. 1a that stool cultures are not negative.

Second, Gamma-proteobacteria are still present, some Firmicutes are also remaining, so there is no complete microbiota elimination. The latter was observed using other protocols of antibiotic cocktail.

Third, this protocol is not the "most commonly used". The one originally devised by Rakoff-Nahoum S et al. (Cell 2004) has been used by many more studies (probably several hundreds), which is not surprising because it's also an easier one to implement.

Finally, the effect of AIMR on glucose homeostasis is not restricted to the protocol described herein but has been recently reported using a mix of antibiotics or individual antibiotics in drinking water (Rodrigues et al. Frontiers Microb 2017 –should be added).

Of note, gamma-proteobacteria are not a phylum.

3) increased glycolysis has been reported in the gut after microbiota introduction into germfree mice (PMID 22722618). This also seems to contradict the results of the current manuscript bringing up again the possibility of confounding effects of antibiotics or some other variables.

In the paper cited by Reviewer #1, the authors investigate the effects of conventionalization of a germ-free mouse on gene expression in the **jejunum**. The authors found that with conventionalization, the jejunum upregulated glycolysis transcriptional pathways. Implicit in the Reviewer's argument is that the jejunum and cecum/colon are similar organs. However, there is decades of research that show that these are clearly two different organs with different physiological and functional properties (see Comprehensive Physiology by APS, Supplement 19;

<http://www.comprehensivephysiology.com/WileyCDA/Section/id-420623.html>) The most salient difference is that SCFAs are the preferred substrate for colonocytes, where butyrate is metabolized in preference to glucose. However, in the intestinal epithelial cells (e.g. in the jejunum), the primary substrate is glucose. Although we cannot comment on the effects of microbiome depletion (or reconstitution) on jejunal transcription, our findings that the upregulation of glycolysis and downregulation of fatty acid oxidation in colonocytes occurs in a setting of the luminal depletion of its preferred substrate, butyrate, does not contradict this excellent article, rather it complements it.

To address the Reviewer's concern, we have made the following change in the discussion:

“This may be a unique process to the colon, since in microbiome depleted states, the jejunum, which does not primarily use SCFAs for energy, increases glycolysis.”

Although difference between small and large intestine is a reasonable argument, the added sentence is incorrect because INCREASED glycolysis was detected AFTER colonization of germfree mice with microbiota (PMID 22722618), thus there is a DOWNREGULATION of glycolysis in GERMFREE small intestine compared to normal state whereas herein it's UP in antibiotic-treated colons.

4) the main claim of the gut as a 'glucose sink' is not sufficiently corroborated. It could be directly compared the measurements of tissue glucose levels between control, AIMD and germfree mice.

We agree with the reviewer that this is conjecture. We have removed this term from the abstract.

5) how can downregulation of gut gluconeogenesis suggested here be reconciled with studies showing that intestinal gluconeogenesis is important for improved systemic glucose tolerance (work by Mithieux lab and others)?

Mithieux lab have shown that **jejunal** gluconeogenesis is important for systemic glucose tolerance. We have not studied the jejunum as part of this manuscript and hence cannot comment on whether intestinal gluconeogenesis is affected by AIMD. As noted before, the jejunum and cecum/colon have different physiological characteristics and different metabolic machinery. Hence, we cannot assume that their response to AIMD will be similar.

6) direct effect of antibiotics on mouse cells cannot be excluded. For example, authors observed a discrepancy between AIMD and GF mice (figs. 4, S4). Can those be related to mitochondrial and other metabolic deficiencies intestinal epithelial or other cells as previously reported (ref.6)? Comparison to GF is quite limited in this study. In particular, in the discussion there is a statement that switch to anaerobic glycolysis cannot be attributed to mitochondrial deficit. Why? With the defect in mitochondrial oxidative phosphorylation, cells are trying to compensate energetic demands by boosting glycolysis.

The Reviewer brings up an excellent point. Morgun, et al. (ref. 6) found that both antibiotics and proteobacteria independently can negatively affect mitochondrial function. Hence, a defect in mitochondrial oxidative phosphorylation can contribute to the observed physiology, and we have revised the section as the Reviewer has recommended. Implicit in the Reviewer's comment is a desire for more comparisons between AIMD and GF mice physiologically. However, that is not the goal of our study. We aimed to assess the metabolic effects of AIMD to understand how it may be affect metabolic experiments performed in these mice.

To address the Reviewer's concern, we have made the following change in the discussion:

“Previous studies which studied the effects of antibiotic administration in GF mice and transplantation of antibiotic-resistant bacteria in GF mice, show that both conditions decrease active mitochondria and inhibit mitochondrial gene expression, which could be contributing to increased epithelial cell death.⁶ We have observed a similar phenomenon where mitochondrial number and gene expression are decreased in AIMD mice. This could also be contributing to increased anaerobic glycolysis observed in AIMD mice.”

7) This work represents a rather extensive description of an AIMD model, which would be not feasible to implement for most labs (gavage 2 times a day every day for 30 days!). More importantly, it is not justified why such protocol is needed, considering enormous number of studies that have been using simpler protocols (drinking water) for more than a decade.

Please see the response to Reviewer #1, Comment (2). Ref 22 clearly shows that most protocols that are employed by researchers, including the ones that the reviewer suggests, do not achieve microbiome depletion. After publication of Ref 22, AIMD protocols have been standardized to avoid confusion and results that have not been replicated. As mentioned in comment (2), AIMD is distinct from antibiotic treatment. It is the responsibility of each investigator to show that their AIMD protocol actually achieves microbiome depletion. We have shown in Fig 1 and S1 that we have achieved microbiome depletion with our protocol. Our own work, and those published by others show that treatment with anything less than twice daily dosing of at least three broad spectrum antibiotics does not achieve microbiome depletion. Hence, studies that do not use these rigorous antibiotic regimens should be interpreted carefully since, although they have decreased microbiome diversity, they have not attained microbiome depletion per se. Finally, a gavage of 30 days is not necessary to observe the metabolic effects noted. In fact, experimental replicates for shorter duration (i.e. 13 days) found similar metabolic effects. Our methods section has now been updated to reflect this point.

8) regarding the microbiome analysis (fig. 1): why showing together Bacteroidetes & Firmicutes? Which genera/species of proteobacteria were detected in AIMD mice?

The goal of AIMD protocol is to deplete the major phyla of the microbiome (i.e. Firmicutes and Bacteroidetes). Fig 1B and 1C show that in the most concise manner. If the reviewer is interested in specific phyla levels, that is already depicted in Fig 1D and Fig S1F. The particular proteobacteria that survive the AIMD protocol are listed in Table S1.

9) What if corrected by decreased overall absolute abundance, would amount of proteobacteria differ between groups, i.e. absolute amount per g of stool?

This is an interesting question and one that we are curious about as well. We can only speculate based on the available data. Fig S1B shows that cecal DNA decreased 20-fold between AIMD and Vehicle-treated mice. In contrast, we see the proportion of proteobacteria increase 10,000-fold. If 16S is proportionally the same in the cecal DNA extracted from both groups (which is unlikely to be a correct assumption), it is likely that proteobacteria are flourishing in the microbiome depleted environment perhaps due to a lack of competition. However, since this is purely speculative, we have not added this information to the manuscript.

10) cecal weight (and likely other parts of distal gut) is increased in almost 2 g in AIMD mice but body weight did not differ from controls. Something is missing. Decreased leptin levels (fig4) indicate that there might be less fat.

This concern was shared with Reviewer #3. We are confident of the body composition data that we have reported for reasons. First, we would like to point out that the body composition data has been replicated (Fig 2H and Fig S2C & D). To address this important concern, we took a closer look at our body composition data. We noted that although the weight of the mice during the experiment was not significantly different (Fig 2F and S2A), on the day of body composition assessment the weight of the vehicle-treated mice was non-significantly less than that of the AIMD mice. We attribute this difference to the fact that weekly weights were done while the animal had ad libitum access to food, but on the day of body composition testing, the mice were fasted. Upon dissection of the mice, we noted that the ceca of AIMD mice were full of watery stool but did not note any difference in subcutaneous or visceral fat. When we reanalyzed the body composition data (please see new Fig 2H), we noted that the non-fat/non-lean mass

compartment was significantly different between the two mouse groups. We attribute this difference to differences in cecal content.

Without performing more detailed analysis of the white adipose tissue, it will be difficult to know what is causing the decrease in circulating leptin. We speculate that there are two reasons for the decreased serum leptin. First, besides adiposity, leptin is released in a fed state (i.e. high circulating glucose) and increased serum insulin. Although fasting insulin is similar between AIMD and vehicle-treated mice, there is a significant drop between fasting and fed insulin in AIMD mice, who are also hypoglycemic, both of which could explain decreased leptin levels in AIMD mice. Another potential reason is the high levels of GLP-1 may inhibit leptin release. Leptin is known to be a potent stimulator of GLP-1 secretion from L-cells (PMID 12540594; 25833771). It is reasonable to speculate that since high GLP-1 is a potent appetite suppressor, decrease leptin levels help regulate food/energy intake.

To address this comment, we have split our body composition graphs into three separate ones showing differences between lean, fat, and non-fat/non-lean mass. In addition, we have modified the discussion to include:

“Furthermore, AIMD induces decreased fasting serum leptin levels without a change in body adiposity. Besides body adiposity, a fed state (as determined with high circulating glucose) and increased serum insulin can both increase leptin release. Although fasting insulin is similar between AIMD and vehicle-treated mice, there is a significantly more precipitous drop between fasting and fed insulin in AIMD mice who are also hypoglycemic, both of which could explain decreased leptin in these mice. In addition, GLP-1 is known to affect satiety and high levels decrease food intake as we have observed in some of our cohorts. High potential levels of GLP-1 may inhibit leptin release. Leptin is known to be a potent stimulator of GLP-1 secretion from L-cells.^{29, 30} Because high levels of serum GLP-1 is a potent appetite suppressor in our AIMD mice, decrease in serum leptin levels may be a response to help regulate food/energy intake. Hence, the relationship between GLP-1 on adipose tissue in general, and leptin release in particular should be more closely investigated.”

11) regarding fig. 3: not all primary BA were lower, some were higher. All abbreviations of figs. 3B,C need to be explained. fig 3C is not the best representation. Why make fig. S3? It can be combined with main fig 3.

We agree with the reviewer that our statement was confusing and have corrected it. The abbreviations for all the bile acids are now in the figure legend. We have removed Fig S3 as requested by Reviewer 2.

12) regarding transcriptomic data: it would be important to know if AIMD transcriptomes are very different from GF or not in order to address the concern of the effects of antibiotics on mammalian tissues. Also, whether immune alterations can be attributed to antibiotic-resistant microbe outgrowth.

The goal of our study was to determine the metabolic effects of AIMD, particularly on glucose homeostasis. Hence, a comparison between AIMD and GF mice is outside the scope of this article. We would like to point out that there is already a very well-written article on gene expression differences between AIMD and GF mice (ref 6), hence performing these experiments would not be novel nor help us attain a better understanding of AIMD effects on metabolism.

13) levels of fasting insulin might be actually different in fig S4a at fasted state with means ~1.5 vs ~0.5. Alas, this was not significant with the p-value for this comparison was 0.12.

14) was the effect specific to TCA? TbmCA was also increased in AMD mice.

We chose TCA since it was remarkably different between AIMD and Vehicle treated mice (7.4-fold difference). The effect of TbmCA was far smaller (3.6-fold difference). Furthermore, the effects of TbmCA on host metabolism have been investigated (Ref. 5) whereas the effects of

TCA are somewhat unexpected. See also response to Reviewer #3, Comment #7. We have been inspired by these results to investigate the effects of bile acids on glucose homeostasis much more closely. Hence, it is outside of the scope of this study and we will have more information on the matter in the future.

15) Would the phenotype be the same after oral GTT?

We have shown an OGTT in Fig 4I. AIMD mice still have a lower glycemic response in a OGTT compared to vehicle treated mice.

16) several lipid metabolism genes show increased expression in AIMD mice (fig.6a), e.g ApoB, ApoA4, Hsd17b12 etc. What's the interpretation of this?

It is difficult to know why these genes are upregulated without further exhaustive experiments that are outside the scope of this manuscript. The reason why some of the genes are upregulated is likely due to multifactorial reasons. For example, ApoB, a protein responsible for shuttling fat, may be upregulated as a compensatory mechanism since the colonocyte is searching for more fat. ApoA4 has antioxidant effects that likely help the colonocyte with damaged mitochondria. We do not wish to include these speculations in our manuscript since we do not have evidence to support them and they do not contribute to the central aim of the paper.

17) using t-test might not be appropriate if the data did not have normal distributions, which might the case at several instances.

We assumed that circulating hormones/cytokines are normally distributed which is supported by a multitude of previous studies. Comparing the data using non-parametric comparisons (i.e. Mann Whitney U test) did not change any of the conclusions and in most cases further reduced the p-values. Certain measures for which we were unsure (e.g. bile acid pool, short-chain fatty acids) we used non-parametric comparisons (e.g. Figure 3).

18) abstract needs to mention analyses of microbiota

We have included information on the effects of AIMD on the composition of the microbiome.

19) it is not stated how many times the experiments were repeated and although stated, data are not shown for different days of duration of the protocol giving "identical" results.

We now include in our methods information about the number of experiments, the length of each experiment and which samples were used for each study. The experiments were repeated four times with twice-a-day dosing, and once with once-a-day dosing. The latter experiment did not achieve microbiome depletion and those results were excluded. The remaining four times were done at the following lengths: 30 days, 19 days, and 13 days (twice). We have mentioned lengths of the experiment for most figures (e.g. Fig 2), but will include more details in the methods due to the reviewer's request.

For clarity, please make sure that each figure legend contains this info.

- Line 108: cannot state "the most effective" since no direct comparison was made with other protocols and some bacteria still remain, which is similar to antibiotic cocktail treatments used by others.

Reviewers' comments:

Reviewer #1 (Remarks to the Author):

please see an attached file

Reviewer #2 (Remarks to the Author):

Compared to the previous version, the authors have markedly improved the manuscript, taking into account a large part of the concerns raised by the reviewers. Particularly, they have improved presentation and analysis of the microbiome results, clarified protocols and methods, and precised GLP1 and leptin results. They also better explain the scope of the paper and moderate the "glucose sink" hypothesis which was not sufficiently evidenced. Moreover, the authors supplemented the manuscript with the different corrections proposed by the reviewers and modified the manuscript to answer several comments of the reviewers.

These additions and corrections clearly increase the significance of the results and the quality of the paper.

However, some concerns persist regarding interpretation of bile acids and SCFA results. Maybe more importantly, the lack of information regarding jejunum gene expression and gut barrier function prevents a global interpretation of the obtained results.

Finally, there is still a major problem regarding stool culturing. Culturing stool on agar should only allow the culturing of Enterobacteria (belonging to the Proteobacteria phylum) (Firmicutes and Bacteroidetes, the more abundant bacteria, do not grow on this medium, particularly if anaerobiosis is not perfect). Therefore, AIMD should lead to increased colony numbers if Proteobacteria flourish (proportion increased 10,000-fold) as mentioned in the response to reviewer 1.

Reviewer #3 (Remarks to the Author):

The authors greatly improved the content of their manuscript based on carefully following the provided feedback. By providing the details of their experimental design, properly citing the previous literature, and performing the additional experiments, now their manuscript is suitable for your journal. At last, to clarify their insistence and the mechanism of GLP-1 secretion on the difference of bile acids and short-chain fatty acids, TGR5KO and GPR43KO mice are useful. However, I will leave this suggestion to the judgment of editor.

RESPONSE TO REVIEWERS' COMMENTS

Reviewer #1 (Remarks to the Author):

We appreciate the constructive feedback from Reviewer #1. These comments have helped us to better clarify the definition of AIMD to place our research in better context of the existing literature. Furthermore, we have better explained why we have chosen to use gavage instead of antibiotic administration by water. Overall, we hope that any remaining concerns of the reviewer have been fully addressed in this latest revision.

1) [In response to Wichman et al. 2013 – ref 3 – similarities between AIMD mice in our paper and germ-free mice in the Wichman paper] Agree but not sure why Fig. 3A is mentioned since it doesn't show germfree mice, only 4C does.

It appears that the Reviewer is referring to 3A in the Wichmann paper as opposed to the 3A in our paper, which suggests that we have been unclear in our response. To clarify, we have prepared the following table showing the complementary findings between antibiotic treated mice in the Wichmann paper to those in our manuscript:

Complementary Observation in Antibiotic-treated mice	Figure in Wichman, et al. (2013) [ref. 3]	Figure in Our Manuscript	Additional information provided in our manuscript
Antibiotic treatment leads to decrease in luminal SCFAs	Figure 2G	Figure 3A	We provide breakdown of the levels of the different SCFAs
Antibiotic treatment leads to increase in proximal colon Gcg mRNA levels	Figure 2I	Figure 4C	We provide Pcsk7 expression which further suggests GLP-1 production

2) [In response to our comment that gut motility is unlikely to be affecting glucose homeostasis] Ok but please add that other factors like lower gut motility might be contributing.

To address the Reviewer's concern, we have added the following sentence to the discussion:

"High levels of GLP-1, as in AIMD mice, likely slow intestinal transit, [3, 29] though the metabolic effects of this are incompletely understood."

3) I still do not agree with the term antibiotic induced microbiome depletion (AIMD), which gives an impression of total elimination of microbiota. But in fact, it is rather an antibiotic-induced microbiome reduction (AIMR). In fact, it is not true what authors responded: "evidence of microbiome depletion (e.g. negative cultures....., whereas most antibiotic protocols do not" . First, it's clear from Fig. 1a that stool cultures are not negative. Second, Gamma-proteobacteria are still present, some Firmicutes are also remaining, so there is no complete microbiota elimination. The latter was observed using other protocols of antibiotic cocktail. Third, this protocol is not the "most commonly used". The one originally devised by Rakoff-Nahoum S et al. (Cell 2004) has been used by many more studies (probably several hundreds), which is not surprising because it's also an easier one to implement. Finally, the effect of AIMR on glucose homeostasis is not restricted to the protocol described herein but has been recently reported using a mix of antibiotics or individual antibiotics in drinking water (Rodrigues et al. Frontiers Microb 2017 –should be added). Of note, gamma-proteobacteria are not a phylum.

The Reviewer brings up two main points which we will address separately:

(a) “I still do not agree with the term antibiotic induced microbiome depletion (AIMD), which gives an impression of total elimination of microbiota. But in fact, it is rather an antibiotic-induced microbiome reduction (AIMR). In fact, it is not true what authors responded: “evidence of microbiome depletion (e.g. negative cultures....., whereas most antibiotic protocols do not” . First, it’s clear from Fig. 1a that stool cultures are not negative.” Second, Gamma-proteobacteria are still present, some Firmicutes are also remaining, so there is no complete microbiota elimination. The latter was observed using other protocols of antibiotic cocktail.

It appears that we have not been clear about what we define as AIMD and we will do a better job of it. We have never claimed total elimination of microbiota (that would perhaps be called Antibiotic Induced Intestinal Sterilization). Reduction and depletion are in fact synonymous, with depletion defined by the Merriam-Webster dictionary as “*reduction in the number or quantity of something.*” We still prefer to use term AIMD since it is consistent with other published work (see for example, the reference the Reviewer has cited – Rakoff-Nahoum S et al. Cell 2004), and the introduction of “AIMR” can be confusing to readers.

To address the reviewer’s comment, we have changed the first sentence of the introduction to the following:

“Recently, antibiotic-induced microbiome depleted (AIMD) mice, where luminal bacteria are reduced with the administration of four or more broad-spectrum antibiotics through gavage or drinking water, have been used in conjunction with, or sometimes instead of, germ-free (GF) mice to investigate the role of the gut microbiome in some pathological conditions.[1, 2, 3]”

(b) “Third, this protocol is not the “most commonly used”. The one originally devised by Rakoff-Nahoum S et al. (Cell 2004) has been used by many more studies (probably several hundreds), which is not surprising because it’s also an easier one to implement. Finally, the effect of AIMR on glucose homeostasis is not restricted to the protocol described herein but has been recently reported using a mix of antibiotics or individual antibiotics in drinking water (Rodrigues et al. Frontiers Microb 2017 –should be added).”

We appreciate these additional references from the Reviewer and have added them to the introduction and discussion of the manuscript. Since we have better defined AIMD (see (a) above), the Reviewer can see that the Rakoff-Nahoum protocol and Rodrigues protocol would fall under our definition of AIMD. This is because these protocols administer four different broad-spectrum antibiotics (ampicillin, vancomycin, neomycin, and metronidazole). We note that only Rakoff-Nahoum showed evidence of depletion whereas Rodrigues did not. As we have already cited in our manuscript and in our previous comments many antibiotic protocols do not use four or more antibiotics and do not demonstrate depletion. These protocols would not be considered AIMD under our definition.

Regarding drug delivery by gavage vs. water administration, we agree with the reviewer that administration through the water would have been much easier and is used by some. However, in our experience gavaging the mice twice daily has been a better method to deliver these antibiotics for a multitude of reasons:

(1) We are assured that each mouse has received the correct amount/dosage of the antibiotics.

(2) It resulted in much less variability and more consistent results that we could easily replicate.

(3) Some of the antibiotics, especially metronidazole, make the drinking water quite bitter. This can cause decreased water intake which could have confounding effects on our metabolic homeostasis measures. In some cases, antibiotic administration through the drinking water has caused mice to die from lack of water consumption (see ref 23).

(4) Often an artificial sweetener is placed in the water to mask the bitterness of metronidazole. In many cases, this is not explicitly reported by the experimenters (e.g. Mukherji, et al. Cell 2013). We now know that artificial sweeteners have significant effects on the gut microbiome and intestinal homeostasis (Suez, et al. Nature 2014). This could explain why our study and many others have not been able to replicate the work of Mukherji, et al.

(5) In this study, we demonstrated replication of more vigorously studied, published protocols of AIMD (ref 23).

(6) Finally, it took a great deal of effort to dissolve all four antibiotics in the same solution for gavage. In fact, the antibiotics would regularly precipitate out after 24-36 hours at 4°C, requiring us to prepare the solutions daily.

Hence given the problems of bitterness of water and precipitation of antibiotics from the water, we were unable to replicate the experimental conditions from Rakoff-Nahoum and hence resorted to using gavage to administer the antibiotics.

(c) “Of note, gamma-proteobacteria are not a phylum.”

Thank you for pointing out this error. We have corrected it.

4) Although difference between small and large intestine is a reasonable argument, the added sentence is incorrect because INCREASED glycolysis was detected AFTER colonization of germfree mice with microbiota (PMID 22722618), thus there is a DOWNREGULATION of glycolysis in GERMFREE small intestine compared to normal state whereas herein it's UP in antibiotic-treated colons.

Thank you for pointing out this error. We have corrected it.

5) For clarity, please make sure that each figure legend contains this info [which experiment the data is being presented from].

As the Reviewer has requested, we have added this information to the figure legends of the data where this information is relevant to interpretation of the results.

6) Line 108: cannot state “the most effective” since no direct comparison was made with other protocols and some bacteria still remain, which is similar to antibiotic cocktail treatments used by others.

We agree with the Reviewer that this statement is not verifiable. We have changed it.

Reviewer #2 (Remarks to the Author):

Compared to the previous version, the authors have markedly improved the manuscript, taking into account a large part of the concerns raised by the reviewers. Particularly, they have improved presentation and analysis of the microbiome results, clarified protocols and methods,

and precised GLP1 and leptin results. They also better explain the scope of the paper and moderate the “glucose sink” hypothesis which was not sufficiently evidenced. Moreover, the authors supplemented the manuscript with the different corrections proposed by the reviewers and modified the manuscript to answer several comments of the reviewers.

These additions and corrections clearly increase the significance of the results and the quality of the paper.

We appreciate Reviewer #2’s recognition that the manuscript has improved substantially, especially with the corrections and comments proposed by the reviewers. Overall, in response to Reviewer #2’s comments, we have addressed whether additional data regarding jejunum and gut permeability would have improved the paper. We have made changes to the manuscript that stress that these factors could also be contributing to metabolic homeostasis of AIMD mice. However, we believe that these important topics would be too different/disjointed from the topic of the manuscript as written.

1) However, some concerns persist regarding interpretation of bile acids and SCFA results.

It is not clear to us what concerns the Reviewer still has about the interpretation of our bile acids and SCFAs results. It is difficult to speculate what the Reviewer is referring to since no concerns about the interpretation of our bile acid and SCFA results were expressed in the initial review. Our results are unequivocal, reproducible, and biologically plausible. AIMD results in (a) depletion of the secondary bile acid pool both in the gut lumen and serum and (b) depletion of total luminal SCFAs, especially butyrate. These results are consistent with what is known about the contribution of the gut microbiome to the luminal environment. Perhaps the Reviewer is referring to the mild increase in isovaleric and caproic acid measures, which comprise a small proportion of the total SCFAs but were higher in AIMD mice. It is currently unclear whether these two SCFAs play any significant role in host metabolism. We agree that these findings are curious and warrant further investigation. Nevertheless, the stated interpretations of our results are based on strict reading of our primary data.

2) Maybe more importantly, the lack of information regarding jejunum gene expression and gut barrier function prevents a global interpretation of the obtained results.

The Reviewer brings up two different topics in this statement that we would like to address separately:

1) *lack of information regarding jejunal gene expression*: We agree with the Reviewer that having additional information about more proximal gut may be interesting and worthy of investigation. Nevertheless, this manuscript is focused on a set of hypotheses that are logically connected. That is, AIMD causes luminal depletion of the SCFAs and changes in the fecal BA pool. Our cecal transcriptomics were performed after we hypothesized that colonocytes will be heavily affected by AIMD due to these luminal changes. The effects of AIMD on jejunal expression was not something we had set out to investigate and its analysis in a post-hoc manner will not do the topic justice nor help us truly understand its role in metabolic homeostasis of AIMD mice. Recently, there is more vigorous investigation of the microbiome of the proximal gut, and we agree with the Reviewer that this would be a fruitful path to pursue after this study. Hence, the effects of AIMD on the proximal gut should be its own independent study, rather than an afterthought to this one.

2) *lack of information regarding gut barrier function*: We agree with the Reviewer that changes in gut barrier function and LPS circulation are interesting and could play an

important role in host metabolism. However, much like jejunal gene expression, it is unclear how these results can be included in this manuscript without appearing superfluous. Furthermore, we have cited an excellent paper that has reported the effects of AIMD on gut barrier function, LPS, and glucose homeostasis (see ref 19). Hence, we are unsure that these experiments would be novel.

Since we agree that the effect of AIMD on jejunal gene expression and gut barrier function will be a worthy investigation if performed separately, we have made the following change to the discussion:

“More recent studies of the microbiome of the proximal GI tract and gut barrier permeability independently play an important role in the host metabolic homeostasis.[19, 30, 32] Focused studies in these two areas may be necessary to appreciate the full effect of this intervention on host metabolic homeostasis.”

3) Finally, there is still a major problem regarding stool culturing. Culturing stool on agar should only allow the culturing of Enterobacteria (belonging to the Proteobacteria phylum) (Firmicutes and Bacteroidetes, the more abundant bacteria, do not grow on this medium, particularly if anaerobiosis is not perfect). Therefore, AIMD should lead to increased colony numbers if Proteobacteria flourish (proportion increased 10,000-fold) as mentioned in the response to reviewer 1.

We appreciate the Reviewer's concern about the stool culture. There are multiple reasons why these concerns may be unwarranted:

(1) Although many members of the Enterobacteria colonize LB agar, and it is primarily used to study lab strains of *E. coli*, LB agar supports the growth of a multitude of organisms and is not limited to Proteobacteria. LB agar is a general medium for microbiology studies and may be used for routine cultivation of any non-fastidious microorganisms. Also, it does not preferentially grow one kind of bacteria over another. (see Chapter 4: Bacterial Culture, Growth, and Development in *Microbiology: An Evolving Science*, 3rd edition, Slonczewski & Foster editors)

(2) We speculated about the 10,000-fold increase based on DNA data. Looking at Fig 1B & 1C would suggest that it is likely lower than that. Furthermore a 10,000-fold increase of a very small number can still be a very small number (e.g. 10,000-fold increase of a bacteria that is $1 \times 10^{-5}\%$ is still a small percentage – 0.1%).

(3) Our results have replicated those reported by previous studies (see ref 23) which also showed depletion of all aerobic bacteria growing on aerobic LB agar media.

Reviewer #3 (Remarks to the Author):

The authors greatly improved the content of their manuscript based on carefully following the provided feedback. By providing the details of their experimental design, properly citing the previous literature, and performing the additional experiments, now their manuscript is suitable for your journal. At last, to clarify their insistence and the mechanism of GLP-1 secretion on the difference of bile acids and short-chain fatty acids, TGR5KO and GPR43KO mice are useful. However, I will leave this suggestion to the judgment of editor.

We appreciate the Reviewer's comments and agree that the transgenic mice can play an important role in understanding the relationship between GLP-1 and bile acid metabolism/signaling. Our future studies will pursue these important questions. Nevertheless, we do not believe they add more to the current study investigating the role of AIMD per se on glucose homeostasis.